# A novel CHCHD10 mutation implicates a Mia40-dependent mitochondrial import deficit in ALS

Carina Lehmer[1], Martin H Schludi[1,2], Linnea Ransom[1], Johanna Greiling[1], Michaela Junghänel[1], Nicole Exner[3], Henrick Riemenschneider[1], Julie van der Zee[4,5], Christine Van Broeckhoven[4,5], Patrick Weydt[6], Michael T Heneka[6,7] & Dieter Edbauer[1,2,*]

## Abstract

*CHCHD10* mutations are linked to amyotrophic lateral sclerosis, but their mode of action is unclear. In a 29-year-old patient with rapid disease progression, we discovered a novel mutation (Q108P) in a conserved residue within the coiled-coil-helix-coiled-coil-helix (CHCH) domain. The aggressive clinical phenotype prompted us to probe its pathogenicity. Unlike the wild-type protein, mitochondrial import of CHCHD10 Q108P was blocked nearly completely resulting in diffuse cytoplasmic localization and reduced stability. Other CHCHD10 variants reported in patients showed impaired mitochondrial import (C122R) or clustering within mitochondria (especially G66V and E127K) often associated with reduced expression. Truncation experiments suggest mitochondrial import of CHCHD10 is mediated by the CHCH domain rather than the proposed N-terminal mitochondrial targeting signal. Knockdown of Mia40, which introduces disulfide bonds into CHCH domain proteins, blocked mitochondrial import of CHCHD10. Overexpression of Mia40 rescued mitochondrial import of CHCHD10 Q108P by enhancing disulfide-bond formation. Since reduction in CHCHD10 inhibits respiration, mutations in its CHCH domain may cause aggressive disease by impairing mitochondrial import. Our data suggest Mia40 upregulation as a potential therapeutic salvage pathway.

**Keywords** amyotrophic lateral sclerosis; CHCHD10; genetics; mitochondria
**Subject Categories** Genetics, Gene Therapy & Genetic Disease; Neuroscience

## Introduction

The recent identification of mutations in *CHCHD10* implicates mitochondrial dysfunction in the pathogenesis of frontotemporal dementia (FTD) and amyotrophic lateral sclerosis (ALS) (Bannwarth *et al*, 2014). CHCHD10 is a small soluble protein with a positively charged N-terminus commonly referred to as a mitochondrial targeting signal (MTS), a central hydrophobic domain and a C-terminal CHCH domain (Perrone *et al*, 2017). Mutations have been reported mainly in the N-terminus and the central hydrophobic domain. However, the exact molecular function of the protein and the effect of these mutations remain unknown. Electron microscopy and biochemical studies suggest that CHCHD10 resides in the mitochondrial contact site and cristae organizing system (MICOS) in the intermembrane space of mitochondria (Bannwarth *et al*, 2014) although that has been recently disputed by others (Burstein *et al*, 2018). In the MICOS complex, CHCHD10 interacts with mitofusin, CHCHD3, and CHCHD6 and it seems to be required for proper packaging of mitochondrial DNA into the nucleoid structures (Genin *et al*, 2016).

Several *CHCHD10* mutations were identified in association studies from ALS/FTD kindreds. The S59L mutation was found in an extended family with variable clinical presentation including classic motoneuron disease, cerebellar ataxia, and frontal lobar cognitive symptoms (Bannwarth *et al*, 2014). Moreover, S59L patients also show ragged-red fiber myopathy indicative of mitochondrial disease. The subsequent identification of a R15L mutation as the causal mutation in several pedigrees of familial ALS by three independent groups corroborated the link to ALS (Johnson *et al*, 2014; Muller *et al*, 2014; Kurzwelly *et al*, 2015), while a more cautious interpretation of these association studies was put forward by others due to incomplete penetrance (van Rheenen *et al*, 2014). Later, a G66V mutation was associated with ALS (Muller *et al*, 2014), the Jokela type of spinal muscular atrophy (Penttila *et al*, 2015), and Charcot-Marie-Tooth disease type 2 (Auranen *et al*, 2015). The typical age-of-onset in these families is in the fifties, and patients show variable clinical presentation and disease duration (1–12 years). Sequencing studies identified several other *CHCHD10* mutations in ALS/FTD cohorts, but lack functional characterization to support pathogenicity (Chaussenot *et al*,

1 German Center for Neurodegenerative Diseases (DZNE) Munich, Munich, Germany
2 Munich Cluster for Systems Neurology (SyNergy), Munich, Germany
3 Biomedical Center (BMC), Biochemistry, Ludwig-Maximilians-Universität München, Munich, Germany
4 Neurodegenerative Brain Diseases Group, Center for Molecular Neurology, VIB, Antwerp, Belgium
5 Laboratory of Neurogenetics, Institute Born-Bunge, University of Antwerp, Antwerp, Belgium
6 Department of Neurodegenerative Diseases and Geriatric Psychiatry, Bonn University Hospital, Bonn, Germany
7 German Center for Neurodegenerative Disease (DZNE) Bonn, Bonn, Germany
*Corresponding author. Tel: +49 89 440046 510; E-mail: dieter.edbauer@dzne.de

2014; Dols-Icardo *et al*, 2015; Zhang *et al*, 2015; Jiao *et al*, 2016; Zhou *et al*, 2017; Blauwendraat *et al*, 2018).

Functional studies of CHCHD10 variants are largely limited to the S59L mutation and have so far not revealed a clear mode of action. Patient fibroblasts with the S59L mutation show an altered mitochondrial network structure, but as mitochondrial fusion is normal, this may be secondary to instability of mitochondrial DNA (Bannwarth *et al*, 2014). Overexpression of human wild-type but not R15L or S59L CHCHD10 rescues the shorter lifespan of *Caenorhabditis elegans* lacking the CHCHD10 homolog *har-1* (Woo *et al*, 2017). The reported inhibition of apoptosis by CHCHD10 S59L (Genin *et al*, 2016) has not been replicated by others (Woo *et al*, 2017) and is difficult to reconcile with a neurodegenerative process. The neuropathological features of *CHCHD10* cases have not been comprehensively characterized, but CHCHD10 was recently linked to synaptic integrity and nuclear retention of TDP-43 (Woo *et al*, 2017), although the latter has not been replicated (Brockmann *et al*, 2018).

Here, we report a novel Q108P mutation in the CHCH domain of CHCHD10 in a very young patient with rapidly progressing classical ALS symptoms, which is in sharp contrast to the slow progression in most *CHCHD10* patients. We show that the Q108P mutation blocks mitochondrial import nearly completely, and examine the mechanism of CHCHD10 mitochondrial import in detail, including rescue strategies. In addition, we analyzed the effect of all other reported missense mutations on protein expression and localization.

# Results

### Identification of CHCHD10 Q108P in an early-onset ALS patient

A 29-year-old male presented with progressive spasticity, starting in the right foot and spreading to the other extremities over 2 years. He reported recurring painful cramps and had recently noticed atrophy in the hand muscles. Neurologic exam revealed spastic tetraparesis, diffuse fasciculations, muscle atrophy in all extremities, hyperactive deep tendon reflexes, a positive Babinski on the right and equivocal on the left. Motor abnormalities were most severe in the right arm. Bulbar, sensory and coordination functions were normal.

The CSF showed slightly elevated proteins (530.2 mg/l) but was otherwise unremarkable. The electrophysiological exam showed chronic and acute neurogenic changes in the cervical, thoracic, and lumbar region.

The family history was unremarkable for neurodegenerative diseases. Both parents are alive and well at 56 and 55 years, respectively. No DNA was available from the parents. Repeat primed PCR detected no *C9orf72* repeat expansion in the index case. Sequencing using a custom panel with genes linked to ALS/FTD and Alzheimer revealed a heterozygous Q108P mutation in CHCHD10, but no mutations in APP, CSF1R, CHMP2B, FUS, GRN, HNRNPA1, HNRNPA2B1, MAPT, MATR3, NEK1, OPTN, PSEN1, PSEN2, SOD1, TARDBP, TBK1, TUBA4A, TREM2, or VCP (see Materials and Methods). Sanger sequencing confirmed a heterozygous Q108P mutation (Fig 1A). Recently, a nonsense variant (Q108*) was reported at the same position in a case with FTD and atypical Parkinson's disease (Perrone *et al*, 2017). The Q108P variant was not found in the 60,706 control exomes curated in the ExAc database, and the

residue is highly conserved between species (Lek *et al*, 2016). Among the species in the ENSEMBL ortholog list, Q108 is fully conserved apart from yeast (asparagine). While most other reported CHCHD10 variants lie in the N-terminal region (e.g., R15L) and the central hydrophobic domain (e.g., S59L and G66V), the novel Q108P mutation is located in the CHCH domain (Fig 1B).

### CHCHD10 Q108P inhibits mitochondrial import nearly completely

CHCHD10 is localized in the intermembrane space of mitochondria, and several pathogenic mutations are near the putative MTS at the N-terminus. Therefore, we asked, how the Q108P mutation affects the localization and function of CHCHD10, and compared it to the R15L mutation, which was independently discovered in several ALS/FTD kindreds. In HeLa cells, the levels of R15L and especially Q108P were reduced in whole cell lysate compared to HA-tagged wild-type CHCHD10 (Fig EV1A). In immunofluorescence experiments, the wild-type protein showed typical mitochondrial staining and colocalization with the mitochondrial marker protein ATP5A1 (Fig 1C). In contrast, CHCHD10 Q108P was diffusely localized all over the cell, without discernible mitochondrial localization, suggesting that this mutation disrupts the mitochondrial import and/or impairs protein folding/stability. While CHCHD10 R15L levels were also reduced, the residual protein still colocalized with mitochondria similar to the wild-type protein. Line scans confirmed the lack of correlation of CHCHD10 Q108P and mitochondrial signal (Fig EV1B).

In addition, biochemical fractionation showed strongly reduced levels of CHCHD10 Q108P in mitochondria compared to wild-type despite similar cytosolic levels in a quantitative analysis (Fig 1D and E). The mitochondrial levels of CHCHD10 R15L consistently appeared lower than for the wild-type protein without reaching statistical significance. A C-terminal anti-CHCHD10 antibody showed comparable expression of exogenous and endogenous CHCHD10, but poorly detected the Q108P mutant protein. Moreover, transfection of the mutant and wild-type CHCHD10 had no effect on the levels and localization of endogenous CHCHD10 arguing against molecular replacement or dominant negative effects. Next, we transduced primary rat hippocampal neurons with lentivirus expressing CHCHD10 variants. Similar to the results in HeLa cells, wild-type and R15L predominantly localized to mitochondria, while Q108P showed diffuse expression in the soma and neurites (Fig 1F).

Next, we analyzed protein stability, because Q108P and R15L showed reduced protein levels compared to wild-type CHCHD10. Therefore, we treated CHCHD10 expressing cells with cycloheximide (CHX) to block protein translation and analyzed the decay of CHCHD10 over a time course of 24 h (Fig EV1C). Quantification confirmed rapid degradation of CHCHD10 Q108P compared to the wild-type (Fig EV1D), which is reflected in an almost fivefold lower half-life time (Fig EV1E). CHCHD10 R15L showed intermediated stability. Together, these data suggest that the Q108P mutation strongly inhibits mitochondrial import leading to enhanced protein degradation in the cytosol.

### CHCHD10 knockdown impairs cellular respiration

Since mitochondrial CHCHD10 levels are likely reduced in the ALS patient with CHCHD10 Q108P mutation, we addressed the

**Figure 1. CHCHD10 Q108P inhibits mitochondrial import.**

A   Genomic DNA of an ALS patient was PCR amplified and subjected to Sanger sequencing. The fluorogram revealed a heterozygous Q108P mutation in exon 3 of *CHCHD10*.

B   Domain structure and known mutations of CHCHD10. R15L is localized in the putative mitochondrial targeting signal ("MTS?"), S59L and G66V in the hydrophobic region and Q108P in the CHCH domain.

C–F Hela cells were transfected (C–E) and primary hippocampal neurons were transduced (F) with HA-tagged CHCHD10 (D10-HA) wild-type (WT), Q108P, or R15L. (C, F) Mitochondrial localization of CHCHD10-HA (D10-HA) was analyzed by co-staining of a mitochondrial ATP synthase subunit (ATP5A1). Cells with similar expression levels were selected for imaging. Scale bars represent 10 μm. (D) Biochemical fractionation of mitochondria and cytosol from transfected HeLa cells. Immunoblot using antibodies against HA, CHCHD10 C-terminus (D10-CT), ATP5A1, and actin. (E) Protein quantification of CHCHD10-HA (D10-HA) in mitochondrial (normalized to ATP5A1) and cytosolic (normalized to actin) fractions. Data are shown as mean ± SD. One-way ANOVA (followed by Dunnett's *post hoc* test against WT): $n$ = 3 biological replicates, mitochondrial WT versus Q108P: *$P$ = 0.0135.

Source data are available online for this figure.

functional role of CHCHD10 focusing on cellular respiration in knockdown experiments using siRNA. CHCHD10 siRNA reduced expression of CHCHD10 mRNA and protein detected by quantitative RT–PCR and immunoblotting in HeLa cells compared to control siRNA (Fig 2A). Using the Seahorse analyzer, we quantified cellular respiration upon CHCHD10 knockdown in HeLa cells. CHCHD10 knockdown cells showed reduced basal respiration and also reduced maximal respiration upon uncoupling with FCCP, resulting in a lower spare respiratory capacity (Fig 2B and C).

Next, we used CRISPR/Cas9 to introduce a frameshift in CHCHD10 in haploid HAP1 cells near Q108. Deletion of 11 base pairs led to a premature stop codon resulting in the deletion of amino acids 110–142 (p.Leu110HisfsTer5, here called

D10 fs). The frame shift caused significant reduction in the CHCHD10 mRNA through nonsense-mediated decay (Fig 2D). While a C-terminal CHCHD10 antibody detected no full-length protein in the edited cells, an N-terminal antibody still detected low levels of truncated CHCHD10 (Fig 2D). D10 fs cells showed reduced spare respiratory capacity (Fig 2E and F), which is consistent with the knockdown data in HeLa cells (Fig 2B and C).

Since primary cells of the Q108P patients were unfortunately not available, we analyzed lymphoblasts from an FTD patient with a heterozygous Q108* mutation (Perrone *et al*, 2017). Consistent with the reported nonsense-mediated decay of the mutant allele and the findings from the very similar CHCHD10 frame shift allele in HAP1

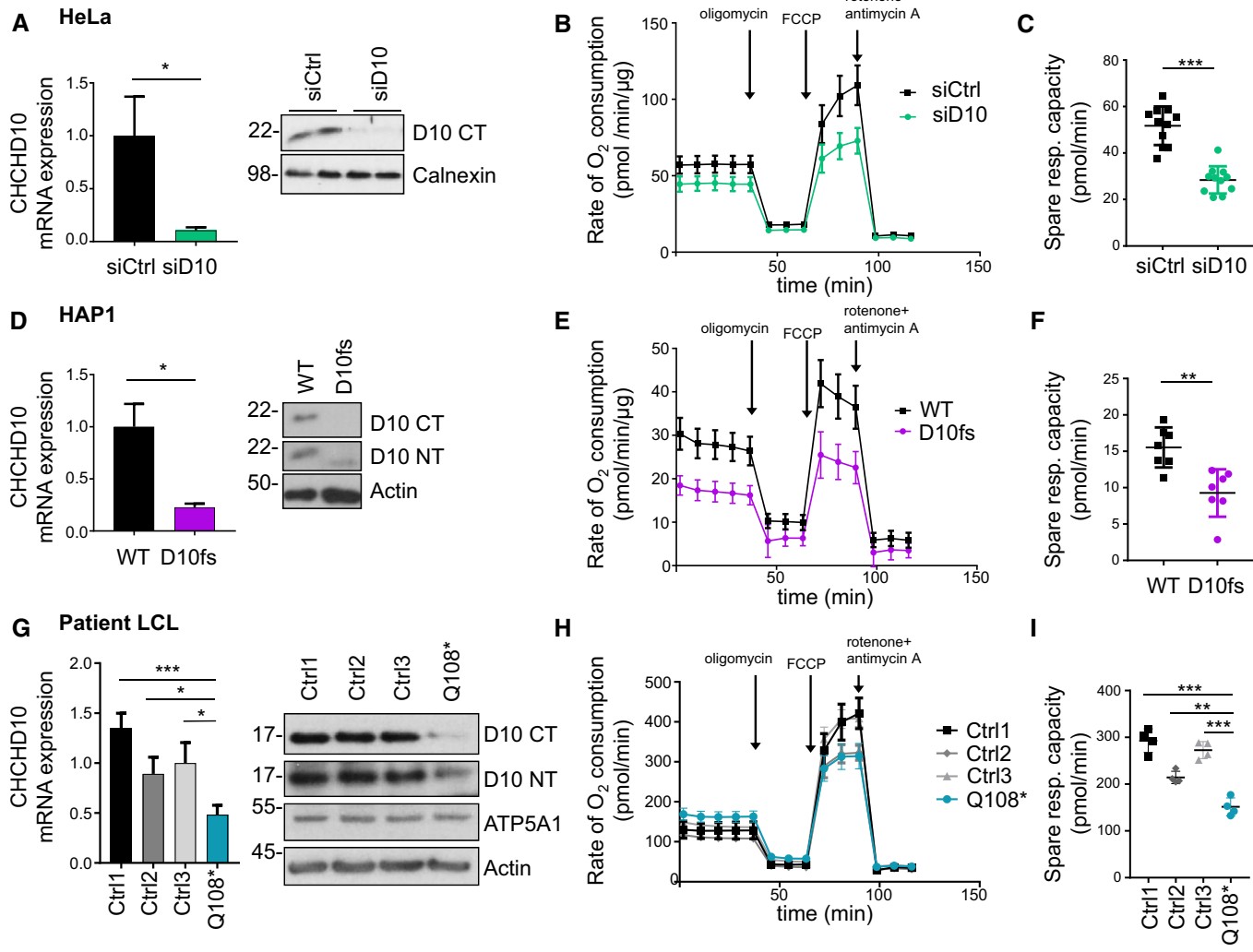

**Figure 2. Partial loss of CHCHD10 reduces spare respiratory capacity.**

A–C   HeLa cells were transfected with siRNA targeting CHCHD10 (siD10) or control (siCtrl). (A) Quantitative RT–PCR and immunoblotting (using a C-terminal antibody) show CHCHD10 knockdown. mRNA levels were normalized to *GAPDH* and *B2M* mRNA. Data are shown as mean ± SD. Welch's *t*-test was used for statistical analysis: *n* = 3 biological replicates, *\*P* = 0.0102. (B, C) Mitochondrial respiration was quantified in real-time using the Seahorse extracellular flux analyzer. The oxygen consumption rate was measured in pmol $O_2$ per minute and normalized to total protein concentration. After measuring basal respiration, oligomycin was added to inhibit ATP synthase (proton leak), followed by the uncoupling agent FCCP (maximal respiration) and antimycin A/rotenone (non-mitochondrial oxygen consumption). Statistical analysis was done for the spare respiratory capacity (difference of maximal and basal respiration). Data are shown as mean ± SD. *T*-test: *n* = 11 biological replicates, *\*\*\*P* < 0.0001.

D–F   CHCHD10 inactivation in haploid HAP1 cells using CRISPR/Cas9 leading to a premature stop codon (p.Leu110HisfsTer5, henceforth abbreviated as D10 fs). (D) Quantitative RT–PCR and immunoblotting (using C- and N-terminal antibodies) show strong reduction of CHCHD10 mRNA expression and loss of full-length protein in D10 fs cells. mRNA levels were normalized to *GAPDH* and *B2M* mRNA. Data are shown as mean ± SD. Welch's *t*-test was used for statistical analysis: *n* = 3 technical replicates, *\*P* = 0.0125. (E, F) Mitochondrial respiration was analyzed as in (B, C). Statistical analysis was done for spare respiratory capacity (difference of maximal and basal respiration). Data are shown as mean ± SD. *T*-Test: *n* = 7 technical replicates, *\*\*P* = 0.0022. A representative experiment of several experiments is shown.

G–I   Lymphoblasts from an FTD patient with a Q108* mutation were compared to three control cases with wild-type CHCHD10. (G) Quantitative RT–PCR and immunoblotting (using C- and N-terminal antibodies) show both reduced CHCHD10 mRNA expression and 50% CHCHD10 protein in Q108* patient cells. mRNA levels were normalized to *GAPDH* and *B2M* mRNA. Data are shown as mean ± SD. One-way ANOVA (followed by Dunnett's *post hoc* test against Q108*) was used for statistical analysis: *n* = 3 technical replicates, Q108* versus Ctrl1: *\*\*\*P* = 0.0004, Q108* versus Ctrl2: *\*P* = 0.0338, Q108* versus Ctrl3: *\*P* = 0.0105. (H, I) Mitochondrial respiration was analyzed 1 h after plating an equal number of lymphoblasts. Statistical analysis was done for spare respiratory capacity (difference of maximal and basal respiration). Data are shown as mean ± SD. One of two independent experiments with similar results was analyzed by one-way ANOVA (followed by Dunnett's *post hoc* test against Q108*): *n* = 4 technical replicates, Q108* versus Ctrl1: *\*\*\*P* = 0.0001, Q108* versus Ctrl2: *\*\*P* = 0.0017, Q108* versus Ctrl3: *\*\*\*P* = 0.0001.

cells, Q108* lymphoblasts show reduced CHCHD10 mRNA and protein levels compared to lymphoblasts from controls with wild-type CHCHD10 (Fig 2G). Reduced CHCHD10 expression in these cells is associated with a reduced spare respiratory capacity compared to the three control lines with wild-type CHCHD10 coding sequence (Fig 2H and I).

Thus, reduced mitochondrial import of CHCHD10 Q108P may decrease mitochondrial function in the early-onset ALS case with only one intact allele.

### The CHCH domain is critical for mitochondrial import

In the current literature, the N-terminus of CHCHD10 is widely referred to as a MTS due to the presence of four interspaced arginine residues. To decipher the contribution of the respective domain to the mitochondrial import mechanism of CHCHD10, we generated truncated CHCHD10 expression constructs and analyzed the mutant proteins by immunofluorescence and biochemical fractionation (Fig 3A–C). Similar to the R15L mutation, truncation of the predicted N-terminal MTS (ΔNT, aa 1–16) had little effect on the mitochondrial import. Deleting the C-terminal CHCH domain (ΔCHCH, aa Δ92–142) strongly reduced protein levels and

prevented mitochondrial import nearly completely. Importantly, the Q108* patient variant inhibited mitochondrial import like the Q108P mutation. Both CHCHD10 ΔCHCH and Q108* proteins were retained in the cytosolic fraction, confirming that an intact CHCH domain is necessary for mitochondrial import of CHCHD10 (Fig 3B and C). Deleting the N-terminus from the Q108P did not further impair mitochondrial import arguing for a dominant role of the CHCH domain (Fig 3B and C).

To determine which domains of CHCHD10 are sufficient for mitochondrial import, we fused the N-terminus (NT-GFP, amino acids 1–33) or the C-terminus (CHCH-GFP and GFP-CHCH, amino acids 88–142) to GFP. While conventional MTS is widely used in fluorescent mitochondrial reporters, the predicted MTS of CHCHD10 was not sufficient for mitochondrial import when fused to GFP (Fig 3D). Unexpectedly, the CHCH domain fused to either the N- or C-terminus of GFP also failed to drive mitochondrial import.

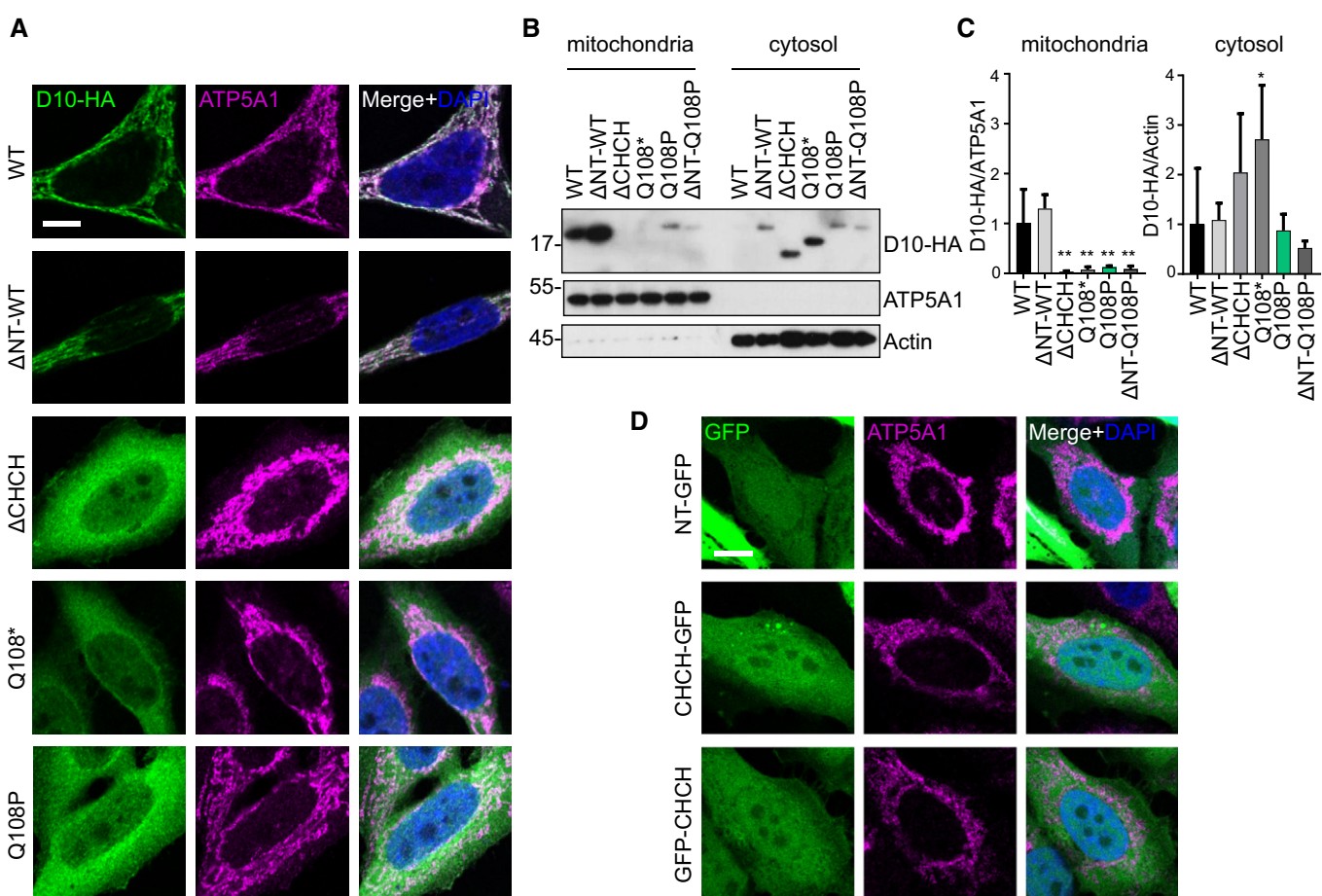

**Figure 3. The CHCH domain is necessary for mitochondrial import of CHCHD10.**

HeLa cells were transfected with the indicated CHCHD10 variants (D10-HA) and GFP-fusion proteins.

A–D  (A, D) Double immunofluorescence using ATP5A1 as a mitochondrial marker protein. Cells with similar expression level are shown. Scale bars represent 10 μm. (B, C) Representative immunoblot of biochemical fractionation of mitochondria and cytosol using antibodies against HA, ATP5A1, and actin followed by quantitative analysis of the respective CHCHD10 truncation mutant. Levels of HA-tagged CHCHD10 (D10-HA) were either normalized to ATP5A1 (for mitochondria) or actin (for cytosol). Data are shown as mean ± SD. One-way ANOVA (followed by Dunnett's *post hoc* test against WT): *n* = 3–4 biological replicates, mitochondrial: WT versus ΔCHCH **P* = 0.0013, WT versus Q108* **P* = 0.0019, WT versus Q108P **P* = 0.0030, WT versus ΔNT-Q108P **P* = 0.0020; Cytosolic: WT versus Q108* *P* = 0.0428.

Source data are available online for this figure.

However, fusing GFP to the N- or C-terminus of full-length CHCHD10 also blocked mitochondrial import of wild-type CHCHD10 (data not shown), indicating that the CHCH domain-mediated import mechanism may not be compatible with large proteins, which unfortunately precludes definite interpretation of this experiment. The truncation experiments show that mitochondrial import of CHCHD10 is predominantly driven by the CHCH domain.

## Mutations in the hydrophobic region and the CHCH domain affect subcellular CHCHD10 distribution

To test, whether impaired mitochondrial import is a common pathomechanism, we examined steady state protein levels and localization of all reported missense CHCHD10 variants (Bannwarth et al, 2014; Ajroud-Driss et al, 2015; Dols-Icardo et al, 2015; Jiao et al, 2016; Perrone et al, 2017; Zhou et al, 2017). In public exome sequencing data from ~ 2,000 ALS patients (ALSdb, Cirulli et al, 2015), we discovered two additional CHCHD10 mutations in the CHCH domain that are rare in the ExAc database (Lek et al, 2016). One case had a heterozygous mutation of an essential cysteine (C122R), and one case had a charge-altering mutation in a highly conserved residue (E127K) within the CHCH domain, suggesting that such mutations significantly contribute to ALS pathogenesis. In addition, this dataset contained novel R6G and G66S variants. To facilitate site-directed mutagenesis of the highly GC-rich sequence, we used a codon-optimized synthetic gene encoding human CHCHD10 (Fig EV2A). The Q108P and R15L mutants had similar effects on expression and localization, although the synthetic gene allowed higher expression levels (Fig EV2B and C).

Importantly, the C122R mutant showed diffuse cytoplasmic localization similar to Q108P (Figs 4A and EV2C). Consistent with previous reports (Woo et al, 2017), CHCHD10 S59L showed small punctate staining in mitochondria in many transfected cells. Even stronger clustering was observed for G66V and E127K in nearly all cells. Other variants in the hydrophobic domain had little (G58R and G66S) or no effect (V57E) on CHCHD10 localization but may have subtle effects on mitochondrial morphology similar to reports for S59L (Bannwarth et al, 2014; Woo et al, 2017). The other variants showed no gross abnormalities in expression level and localization by immunofluorescence (Fig EV3), highlighting the importance of the hydrophobic region and the CHCH domain.

For a more quantitative analysis, we analyzed CHCHD10 protein levels 3 days after transfection (Fig 4B). CHCHD10 P23S, G58R, G66V, Q108P, and C122R levels were significantly reduced compared to wild-type. Surprisingly, expression of the common P34S variant and R6G, R15S, A32D, and A35D was enhanced arguing against pathogenicity. Biochemical fractionation confirmed that C122R strongly inhibits mitochondrial import similarly to Q108P suggesting that disulfide-bond formation in the CHCH domain is critical for mitochondrial import (Fig 4C and D).

## Mia40 mediates mitochondrial import of CHCHD10

To test whether CHCHD10 is imported into mitochondria via the Mia40 redox system similar to other CHCH domain containing proteins, we used siRNA to inhibit this pathway, also including the FAD-linked sulfhydryl oxidase Erv1 and AIFM1. RT–qPCR and immunoblotting confirmed the potency and specificity of all siRNAs

(Fig 5A–C). Strikingly, Mia40 knockdown strongly reduced the levels of endogenous CHCHD10 protein despite unchanged mRNA levels. Knockdown of AIFM1 and Erv1 also seemed to decrease CHCHD10 protein levels slightly, however, without reaching statistical significance (Fig 5C). Immunofluorescence confirmed colocalization of endogenous CHCHD10 with mitochondrial cytochrome c oxidase II (MTCO2; Fig 5D). In contrast to control, Mia40 knockdown strongly reduced overall CHCHD10 levels and prevented mitochondrial targeting. Due to the low CHCHD10 protein levels in Mia40 knockdown, we speculate that CHCHD10 mislocalized to the cytosol is degraded rapidly similar to our findings for Q108P (Fig EV1C–E).

Mia40 mediates import of its substrates by direct binding and disulfide-bond formation, which traps the target proteins in the mitochondria (Peleh et al, 2016). Therefore, we analyzed interaction of CHCHD10 with Mia40 in cotransfected HeLa cells. Co-immunoprecipitation experiments showed interaction of wild-type, Q108P, and R15L CHCHD10 with Mia40, but no interaction with the ΔCHCH construct and only weak interaction with the Q108* construct (Fig EV4A).

To directly probe Mia40-mediated disulfide-bond formation in the CHCH domain, we treated cell extracts with 4-acetamido-4′-maleimidylstilbene-2,2′-disulfonic acid (AMS), which is covalently linked to free thiol-groups and thus leads to slower migration in SDS–PAGE. AMS treatment of non-reduced extracts had no effect on wild-type CHCHD10 migration indicating that all cysteine residues are oxidized under basal conditions (Fig 5E). Prior reduction with DTT increased the apparent molecular weight of wild-type CHCHD10, particularly upon heating samples to 95°C, presumably due to increased reduction efficiency. Similar results were obtained for endogenous CHCHD10 (Fig EV4B). While CHCHD10 Q108P levels were lower under all conditions, heating the CHCHD10 Q108P extracts during DTT treatment had no additional effect on AMS accessibility in contrast to the wild-type. Thus, the Q108P mutant is completely reduced by DTT already at room temperature indicating that the CHCH domain in the Q108P mutant may be misfolded. Moreover, treating CHCHD10 R15L extract with AMS showed results similar to wild-type, suggesting normal formation of disulfide bonds in the intermembrane space of mitochondria in this mutant.

## Mia40 overexpression restores mitochondrial import of CHCHD10 Q108P

Since mitochondrial import of wild-type CHCHD10 depends on the integrity of the Mia40 system, we asked how the patient-derived variants are affected by this pathway. First, we analyzed the impact of Mia40 overexpression on the localization of CHCHD10 Q108P in HeLa cells. Strikingly, Mia40 promoted mitochondrial import of CHCHD10 Q108P (Fig 6A). The rescue of mitochondrial import of CHCHD10 Q108P due to Mia40 overexpression was fully replicated in primary neurons (Fig 6B). Additionally, biochemical fractionation and quantification confirmed that overexpression of Mia40 increased the levels of wild-type, Q108P, and R15L CHCHD10 in isolated mitochondria from HeLa cells (Fig 6C and D, also seen in input of Fig EV4A). Overexpressed Mia40 increased also wild-type and mutant CHCHD10 in the cytosolic fraction, which may be explained by partial

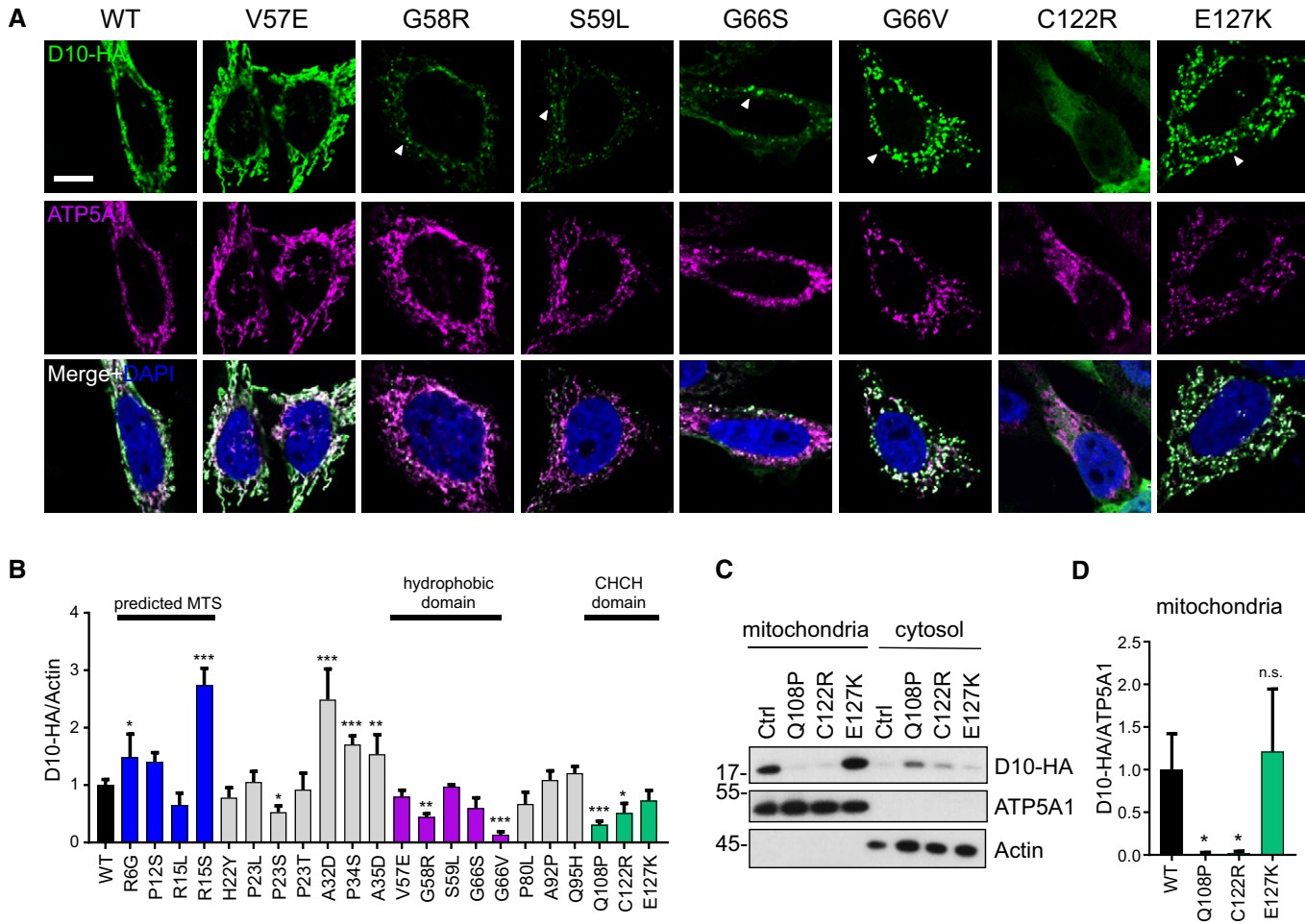

**Figure 4. Differential effect of CHCHD10 patient variants on localization and expression.**

HeLa cells were transfected with HA-tagged CHCHD10 (D10-HA) patient variants.

A Immunofluorescence shows expression pattern of CHCHD10-HA variants compared to the mitochondrial marker ATP5A1. Arrowheads indicate clustering of CHCHD10 within mitochondria. Scale bar represents 10 μm.

B Quantification of CHCHD10 levels from immunoblots of whole cell lysates. Data are shown as mean ± SD. One-way ANOVA (followed by Dunnett's *post hoc* test against WT): *n* = 3–6 biological replicates, WT versus R6G: *\*P* = 0.0145, WT versus R15S: *\*\*\*P* = 0.0001, WT versus P23S: *\*P* = 0.0189, WT versus A32D: *\*\*\*P* = 0.0001, WT versus P34S: *\*\*\*P* = 0.0001, WT versus A35D: *\*\*P* = 0.0044, WT versus G58R: *\*\*P* = 0.0029, WT versus G66V: *\*\*\*P* = 0.0001, WT versus Q108P: *\*\*\*P* = 0.0001, WT versus C122R *\*P* = 0.0146.

C Immunoblot of biochemical fractionation of mitochondria and cytosol from transfected HeLa cells expressing different CHCHD10 patient variants using antibodies against HA, ATP5A1, and actin.

D Quantification of CHCHD10-HA protein level normalized to mitochondrial ATP5A1. Data are shown as mean ± SD. One-way ANOVA (with Dunnett's *post hoc* test against WT): *n* = 4 biological replicates, Mitochondrial CHCHD10 WT versus Q108P: *\*P* = 0.0156, WT versus C122R: *\*P* = 0.0172.

Source data are available online for this figure.

cytosolic localization of excess Mia40 (Fig EV4C). Importantly, Mia40 expression also enhanced CHCHD10 Q108P stability (Fig EV4D and E). Moreover, biochemical analysis of CHCHD10 disulfide-bond formation using AMS treatment confirmed Mia40-induced oxidation and mitochondrial import of Q108P CHCHD10. Without Mia40 overexpression, the CHCHD10 Q108P mutant was poorly expressed (Fig 6E). However, co-expression of Mia40 resulted in higher protein expression and disulfide-bond formation comparable to wild-type CHCHD10, suggesting that oxidation via Mia40 is crucial for the stability and mitochondrial localization of CHCHD10 Q108P. Thus, Mia40 overexpression likely restores mitochondrial import of CHCHD10 Q108P by promoting disulfide-bond formation.

## Discussion

Unusual phenotypes of genetically determined diseases offer an opportunity to explore molecular pathomechanisms. The known CHCHD10 mutations are usually associated with slow progressing forms of late-onset motoneuron disease and frontotemporal dementia. Here, we identified a novel CHCHD10 mutation in a

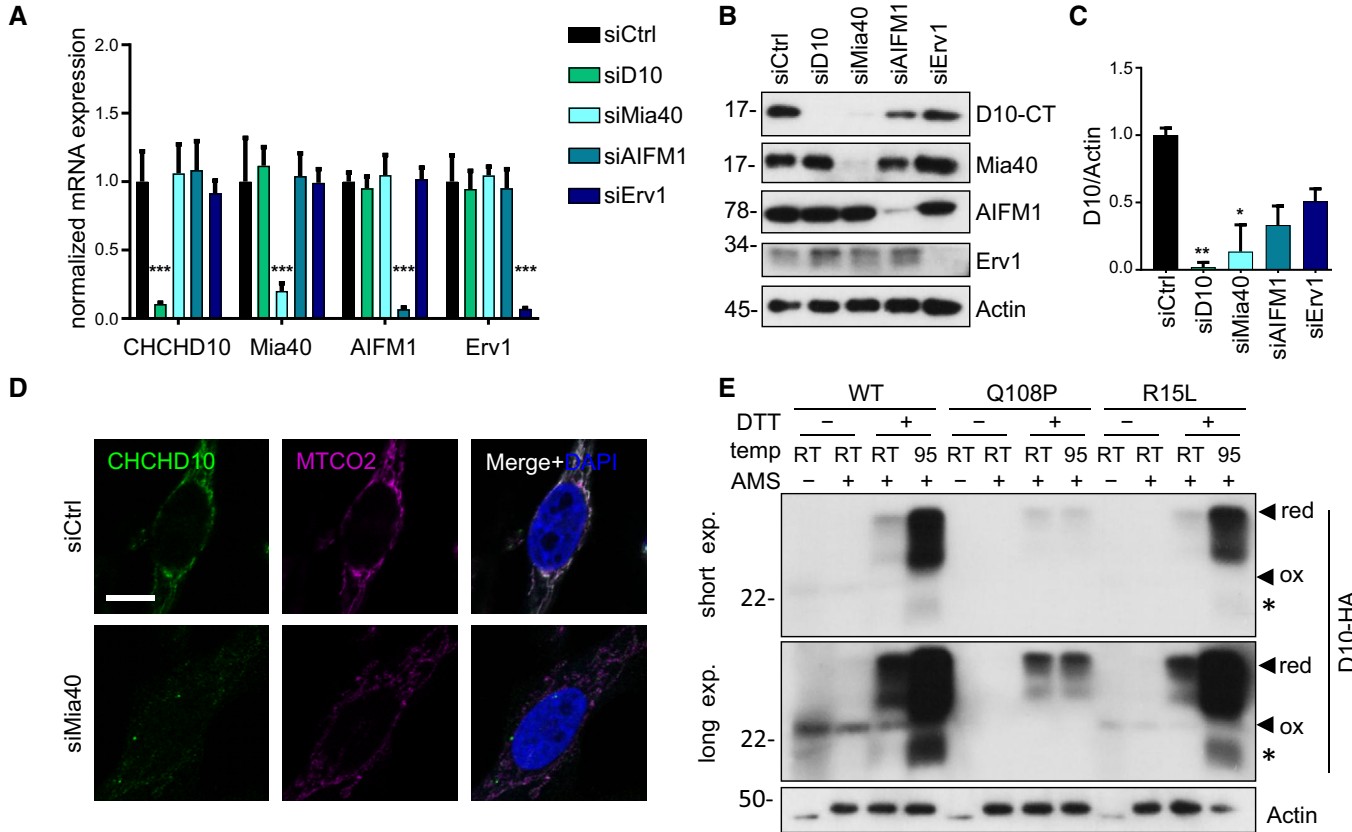

**Figure 5. Mitochondrial import of CHCHD10 depends on Mia40.**

A–D  HeLa cells were transfected with siRNA targeting CHCHD10, Mia40, AIFM1, Erv1, or control (siCtrl). (A) Quantitative RT–PCR confirm specific knockdown of CHCHD10, Mia40, AIFM1, and Erv1. mRNA levels were normalized to *GAPDH* and *B2M* mRNA. Data are shown as mean ± SD. One-way ANOVA (followed by Dunnett's multiple comparisons test against siCtrl) was used for statistical analysis: $n = 4$ biological replicates, siCtrl versus siD10 ***$P = 0.0001$, siCtrl versus siMia40 ***$P = 0.0001$, siCtrl versus siAIFM1 ***$P = 0.0001$, siCtrl versus siErv1 ***$P = 0.0001$. (B) Immunoblots with indicated antibodies in siRNA transfected cells. (C) CHCHD10 protein quantification of siRNA transfected cells normalized to actin. Data are shown as mean ± SD. Kruskal–Wallis test: $n = 4$ biological replicates, siCtrl versus siD10: **$P = 0.0013$, siCtrl versus siMia40: **$P = 0.0136$. (D) Immunostaining of Mia40 knockdown HeLa cells shows overall reduced expression of CHCHD10 compared to control (siCtrl). An antibody against mitochondrially encoded cytochrome c oxidase II (MTCO2) labels mitochondria. Scale bar represents 10 μm.

E  AMS assay to assess disulfide-bond formation in whole cell extracts of HeLa cells transfected with CHCHD10-HA wild-type (WT) and mutants (Q108P, R15L). Extracts were treated with the thiol-reactive cross-linker AMS (10 mM, 37°C, 60 min) with or without prior reduction with DTT and heat denaturation (95°C, 10 min), and subjected to immunoblotting to analyze AMS-induced gel shift from oxidized (ox) to reduced (red) forms of CHCHD10. Note that 95°C treatment has no additional effect on AMS accessibility of CHCHD10 Q108P indicating impaired folding compared to wild-type and R15L. Upper and lower panel show short and long exposure of the same blot, respectively. Asterisk denotes degradation product.

Source data are available online for this figure.

young ALS patient with an aggressive disease course and analyze the consequences for protein function. The Q108P mutation inhibits mitochondrial import of CHCHD10 via the Mia40 system nearly completely. Rescue of mitochondrial import by Mia40 overexpression suggests that Q108P reduces binding affinity to Mia40 and can be compensated for by excess Mia40. In contrast, the common R15L mutation had a much smaller effect on protein levels and subcellular distribution, while several mutations in the hydrophobic domain cause clustering of CHCHD10 within mitochondria. Thus, the strong effect of CHCHD10 Q108P on mitochondrial import may explain the aggressive disease in the mutation carrier and suggests that CHCHD10 is important for mitochondrial respiration in motoneurons during healthy aging.

## Mitochondrial import of CHCHD10 via Mia40

To address the pathogenicity of the novel Q108P variant in CHCHD10, we expressed the mutant protein in HeLa cells and primary hippocampal neurons and noticed diffuse localization all over the cell compared to predominantly mitochondrial localization of the wild-type. Our findings suggest that impaired mitochondrial import is the main pathogenic mechanism for the CHCHD10 Q108P variant and led us to investigate the mitochondrial import mechanisms of wild-type and mutant CHCHD10 in more detail.

Apart from the 13 proteins encoded on the mitochondrial DNA, all other ~ 1,500 mitochondrial proteins are synthesized in the cytosol and require active transport into mitochondria (Wiedemann & Pfanner, 2017). The vast majority of nuclear encoded proteins

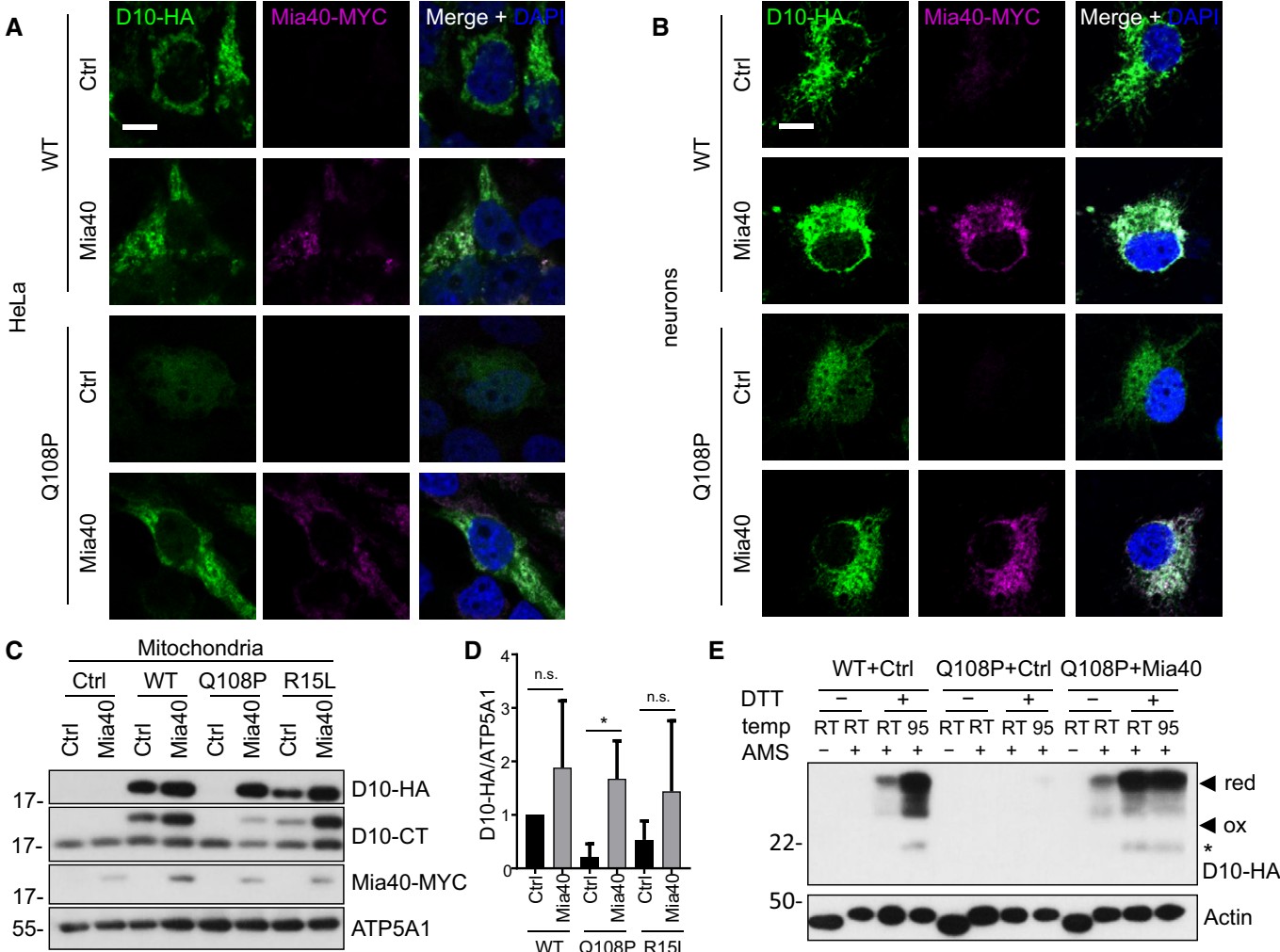

**Figure 6.  Mia40 overexpression rescues CHCHD10 mutants.**

Co-transfection of HeLa cells (A, C, D, E) and co-transduction of primary hippocampal rat neurons (B) with the indicated HA-tagged CHCHD10 (D10-HA) variants and Mia40-MYC or empty vector (Ctrl).

A, B   Immunofluorescence shows colocalization of wild-type CHCHD10 with Mia40. Scale bars represent 10 μm. Overexpression of Mia40 promotes expression and mitochondrial localization of CHCHD10 Q108P.

C, D   Immunoblot and quantification of mitochondrial fraction confirms CHCHD10 stabilization and increased mitochondrial localization upon Mia40 expression compared to empty vector. Quantification normalized to ATP5A1. Data are shown as mean ± SD. Kruskal–Wallis test: *n* = 4 biological replicates. Q108P Ctrl versus Q108P Mia40 **P* = 0.0126.

E   AMS treatment visualizes disulfide-bond formation in CHCHD10 Q108P upon Mia40 expression comparable to wild-type CHCHD10 (with endogenous Mia40 levels). Actin is used as loading control. Note that DTT treatment has no effect on AMS cross-linking of actin, because all its cysteines are reduced in the cytoplasmic environment. Asterisk denotes degradation product.

Source data are available online for this figure.

have to pass through the translocator of the outer membrane (TOM). Distinct machinery directs these proteins further to their final destination in the outer membrane, the intermembrane space, the inner membrane or the matrix, depending on additional sequence motifs. The classical import pathway is triggered by an amphipathic N-terminal MTS recognized by the TOM complex. For CHCHD10, the NCBI annotation and bioinformatic predictions tools (e.g., Psort2 and MitoProt II) suggest the presence of a classical N-terminal MTS with interspaced conserved arginines (amino acids 1–16). So far, the N-terminal region has been interpreted as an MTS in several papers without rigorous experimental validation

(e.g., Perrone *et al*, 2017). Disruption of this putative MTS could potentially explain pathogenicity of the common N-terminal mutations. However, the R15L mutant was still localized to mitochondria and expression levels and stability of CHCHD10 were only slightly reduced, which is consistent with previous colocalization data for this mutant (Woo *et al*, 2017). Other tested N-terminal variants (R6G, P12S) did not reduce expression levels noticeable or even increased expression (R15S). The dramatic reduction of endogenous CHCHD10 levels in Mia40 knockdown cells suggests that cytosolic CHCHD10 lacking the characteristic disulfide bonds is misfolded and rapidly degraded. We therefore cannot exclude that the reduced

expression of R15L is due to slightly less efficient mitochondrial import. Indeed, consistent with other recent reports, CHCHD10 R15L has a shorter half-life time than wild-type (Brockmann *et al*, 2018). The N-terminal arginine-rich sequence may enhance mitochondrial import although it is neither necessary nor sufficient for mitochondrial import by itself. Truncation of the N-terminus did not further impair mitochondrial import in Q108P. The more dramatic effect of the Q108P mutation on mitochondrial import and half-life time may explain the early age of onset in our patient.

Deletion of the whole CHCH domain completely abolished mitochondrial import of CHCHD10. The Mia40 redox system mediates import of proteins with twin $CX_3C$ and $CX_9C$ motifs into the inter-membrane space, including CHCH domain proteins (Mesecke *et al*, 2005). We show that mitochondrial import of CHCHD10 critically depends on Mia40. Strikingly, overexpression of Mia40 promotes import of not only wild-type CHCHD10 but also the Q108P and R15L mutants. In yeast, Mia40 levels are rate limiting for mitochondrial import suggesting it acts as a trans-site receptor for import (Peleh *et al*, 2016). In addition, disulfide-bond formation is impaired in the Q108P mutant, which may be due to disturbed α-helix formation in the CHCH domain because proline is a strong helix breaker (Darshi *et al*, 2012). Interestingly, exome sequencing of ~ 2,000 mostly sporadic ALS cases revealed a mutation (C122R) in one of the critical cysteines in the CHCH domain that also impaired mitochondrial import (ALSdb, Cirulli *et al*, 2015).

Our analysis of all reported missense CHCHD10 variants suggests that mutations within the hydrophobic region (G58R, S59L, G66V, and G66S) might invoke additional pathomechanisms because they still allow mitochondrial targeting but lead to intra-mitochondrial clustering. Surprisingly, a similar localization pattern was observed for the E127K variant, but not for other variants in the CHCH domain.

Recently, partial nuclear localization and transcriptional effects of CHCHD10 and a homologous protein, CHCHD2, have been reported (Aras *et al*, 2015, 2017; Woo *et al*, 2017), particularly under stress conditions such as TDP-43 overexpression or oxidative stress. We detected some nuclear staining (Figs 1B and 3A) for CHCHD10 Q108P and other variants with strongly impaired mitochondrial import, suggesting they might additionally cause a gain of toxic function.

### Relevance of CHCHD10 impairment for ALS/FTD

Mitochondrial dysfunction has long been implicated in the pathogenesis of ALS (Smith *et al*, 2017). ALS-causing mutations in SOD1 inhibit respiration and cause mitochondrial damages (Magrane *et al*, 2009), and poly-Gly-Arg/Pro-Arg translated from the expanded *C9orf72* hexanucleotide repeat induce oxidative stress and disrupt mitochondrial architecture (Lopez-Gonzalez *et al*, 2016). Furthermore, pathogenic OPTN mutations impair mitochondrial clearance by mitophagy (Wong & Holzbaur, 2014). Mitochondrial dysfunction has been linked to other neurodegenerative diseases and may explain the broad clinical symptoms associated with CHCHD10 mutations. Interestingly, we noticed reduced spare respiratory capacity upon CHCHD10 knockdown or CRISPR/Cas9-mediated truncation consistent with findings in patient fibroblasts with CHCHD10 S59L (Genin *et al*, 2016). Importantly, spare respiratory capacity was also reduced in lymphoblasts showing reduced CHCHD10 expression due to nonsense-mediated decay caused by a

Q108* mutation in an FTD patient (Perrone *et al*, 2017). This may impair ATP synthesis in patient motoneurons or muscle and may be accompanied by enhanced formation of damaging reactive oxygen species. Several previous reports of conflicting findings of respiratory function in CHCHD10 cellular models (mutant, knockdown, and overexpression) and the recent finding of impaired respiration in muscle but not in whole brain of homozygous CHCHD10 knockout mice suggest cell type-specific effects are at play (Burstein *et al*, 2018; Straub *et al*, 2018). Interestingly, CHCHD10 knockdown in zebrafish also causes muscle pathology (Brockmann *et al*, 2018). Altered metabolism in muscle may promote to ALS pathogenesis (Loeffler *et al*, 2016). AIFM1, which is required for mitochondrial targeting of Mia40, and thus indirectly of CHCHD10, has been linked to mitochondrial encephalopathy and axonal neuropathy (Ghezzi *et al*, 2010; Rinaldi *et al*, 2012). AIFM1 knockdown appeared to reduced CHCHD10 levels after 3 days (without reaching statistical significance), but longer knockdown may be required for a more severe effect due to its indirect action via Mia40.

Together, our data demonstrate that the Q108P mutation almost completely prevents mitochondrial import and perturbed mitochondrial function may ultimately lead to motoneuron degeneration. The stronger effect of Q108P on mitochondrial import compared to previously characterize pathogenic variants may explain the early onset and aggressive course of ALS in our patient. Our findings have implications for genetic counseling of novel CHCHD10 variants and suggest future therapeutic approaches: (i) Variants in conserved residues of the CHCH domain and nonsense mutations (e.g., the previously reported Q108*) are likely pathogenic. Variants in the hydrophobic region primarily alter CHCHD10 distribution within mitochondria. Thus, analyzing mitochondrial import and clustering within mitochondria may be used to assess pathogenicity of novel variants. (ii) Unless the mutant CHCHD10 causes a toxic gain-of-function phenotype, epigenetic boosting of CHCHD10 expression may rescue haploinsufficiency by increasing expression of the wild-type allele. (iii) It may be possible to pharmacologically activate the Mia40/Erv1 disulfide relay system using small redox-compounds. Boosting Mia40 activity or expression may promote import of mutant and wild-type CHCHD10 and thus restore its function within mitochondrial respiration. Most importantly, our report of a novel aggressive mutation with clear functional consequences strongly supports the genetic linkage of CHCHD10 to ALS/FTD pathogenesis.

# Materials and Methods

### Patient materials, clinical history, and sequencing

All procedures on human subjects were in accordance with the WGA Declaration of Helsinki and the Department of Health and Human Services Belmont Report. The Q108P patient consented to diagnostic DNA testing for ALS mutations. All information was obtained from the hospital files. No experiments were done on the patient or using patient material. Genomic DNA was sequenced with a TruSeq Custom Amplicon kit on a MiSeq (Illumina) according to the protocol from the manufacturer. The custom gene panel covered all exons of CHCHD10, CHMP2B, GRN, MAPT, NEK1, OPTN, PSEN1, PSEN2, SOD1, TARDBP, TBK1, TUBA4A, TREM2 and the exons with known pathogenic mutations of APP (exons 12–15),

CSF1R (exon 13–21), FUS (exon 6, 14, 15), HNRNPA1 (exon 9), HNRNPA2B1 (exon 10), MATR3 (exon 1), VCP (exons 3, 5, 6, 11). The CHCHD10 Q108P mutation was confirmed by Sanger Sequencing of genomic DNA (primers GTGGCCCCAGGTTTGAAAC and CAATCTGGTGTTGTGGTCTGG). Repeat primed PCR for *C9orf72* repeat expansion was performed as described previously (van der Zee *et al*, 2013).

Epstein–Barr virus (EBV)-transformed lymphoblast cell lines were established according to standard procedures for previously reported patients and controls (Perrone *et al*, 2017). All subjects had given informed consent.

### DNA constructs, siRNA, and transfection

CHCHD10 and Mia40/CHCHD4 were amplified from HEK293T cDNA and cloned in the FUW3a lentiviral expression vector containing a C-terminal HA or myc epitope tag. As controls we used the empty vectors containing only the epitope tag. The following CHCHD10 truncations were generated: Δ1–16 (ΔNT), Δ108–142 (Q108*), Δ92–142 (ΔCHCH). Q108P and R15L were introduced by standard mutagenesis. For Figs 4 and EV2, we introduced several patient variants in a codon-optimized synthetic gene with reduced GC-content encoding human CHCHD10. All constructs were sequence verified. We used Silencer Select siRNA targeting human Mia40/CHCHD4 (s43607, Thermo Fisher Scientific), human CHCHD10 (s53406, Thermo Fisher Scientific), human Erv1/Gfer (s5704, Thermo Fisher Scientific), human AIFM1 (s17440, Thermo Fisher Scientific), and the Silencer Select Negative Control No. 1 (#4390844, Thermo Fisher Scientific). HeLa cells were transfected using Lipofectamine 2000 (Thermo Fisher Scientific).

### CRISPR/Cas9 genome editing

HAP1 cells (Horizon Discovery) were transfected with Cas9 (Addgene plasmid #52962) and sgRNA (TCTGAGTGGTGGAA CAGTCC in Addgene plasmid #41824) using Lipofectamine 3000 (Thermo Fisher Scientific). After 12 h, medium was exchanged for 24 h before splitting into selection medium containing 8 μg/ml blasticidin and 400 μg/ml Zeocin. After 3 days, selection medium was removed and cultured for 10–14 days till single cell clones were visible. Individual clones were picked and cultured in 96 wells. For screening, genomic DNA was extracted with the NucleoSpin Tissue 96 well kit (Macherey-Nagel) according to manufacturer's instructions. The region of CHCHD10 targeted by the sgRNA was PCR amplified (GGTTTGAAACGCACCTCCAG and AGGTGCAAGAGGA GGGTTG) using the Q5 High-Fidelity Master Mix (New England Biolabs) and analyzed by Sanger sequencing.

### Antibodies

The following primary antibodies were used: anti-HA (clone 3F10, Thermo Fisher - IF 1:10, WB 1:50), anti-myc (9E10 hybridoma supernatant, WB 1:15, supernatant of clone 9E10, IF 1:200, purified), anti-ATP5A1 (WB 1:1,000, IF 1:250, clone 15H4C4, abcam 14748), anti-CHCHD10 (C-terminal WB 1:500, IF 1:100, abcam 121196), anti-CHCHD10 (N-terminal, WB 1:500, abcam ab124186), anti-MTCO2 (IF 1:100, abcam 3298), anti-actin (WB 1:3,000, clone A5316, Sigma), anti-calnexin (WB 1:7,000, clone SPA-860, Enzo Life

Sciences), anti-CHCHD4 (Mia40, WB: 1:1,000, Proteintech 21090-1-AP) anti-AIF (AIFM1, WB 1:1,000, abcam ab32516), anti-Gfer (Erv1, WB 1:200, Atlas Antibodies HPA041227).

### Cell culture, mitochondrial fractionation

HeLa cells were transfected with plasmids and siRNA using Lipofectamine 2000 (Thermo Fisher Scientific) according to the manufacturer's instructions. Three days after transfection, mitochondria were isolated using the Qproteome Mitochondria Isolation Kit (Qiagen). The cytosolic fraction was precipitated with four volumes of ice-cold acetone and incubated for 15 min on ice. After centrifugation (10 min, 12,000 g, 4°C), the pellet was washed twice with acetone and air dried. The cytosolic pellet and the highly purified mitochondrial pellet were resuspended in RIPA buffer (137 mM NaCl, 20 mM Tris pH 7.5, 0.1% SDS, 10% glycerol, 1% Triton X-100, 0.5% deoxycholate, 2 mM EDTA) containing protease inhibitor cocktails (1:100, Sigma), incubated for 20 min on ice, and sonicated for 10s. Afterward, the protein concentration was determined using BCA assay (Interchim). After adding 4× Laemmli buffer (Bio-Rad) containing 2-mercaptoethanol, samples were denatured (95°C, 10 min) and loaded with the same protein amount on Novex 10–20% Tris-Tricine gels (Life Technologies).

### Protein stability and immunoblotting

For protein stability analysis, HeLa cells were treated 2 days after transfection with 150 μg/ml cycloheximide dissolved in DMSO or DMSO only for 0, 4, 8, and 24 h.

For immunoblotting of the whole cell lysates, cells were lysed in RIPA buffer (137 mM NaCl, 20 mM Tris pH 7.5, 0.1% SDS, 10% glycerol, 1% Triton X-100, 0.5% deoxycholate, 2 mM EDTA) with protease inhibitor cocktails (1:100, Sigma) and incubated on ice (20 min). After centrifugation (18,000 g, 15 min), the supernatant was transferred into a new tube, protein concentration was determined by BCA assay (Interchim), and 4× Laemmli buffer (Bio-Rad) containing 2-mercaptoethanol was added. Samples were denatured at 95°C for 10 min and loaded on Novex 10–20% Tris-Tricine gels (Life Technologies) or 12.5% SDS–PAGE gels.

### Immunoprecipitation

HeLa cells were lysed at 4°C for 20 min in lysis buffer (120 mM NaCl, 1 mM EDTA, 0.5% NP-40, 20 mM Tris–HCL pH 8) supplemented with protease and phosphatase inhibitors and centrifuged at 13,000 g for 10 min. HA-labeled magnetic beads (Thermo Fischer 88836) were washed with 4°C lysis buffer; 5% of the cell lysate was used as an input control and the rest of the cell lysate was incubated at 4°C with HA-labeled beads overnight. Beads were washed three times with 4°C lysis buffer supplemented with protease and phosphatase inhibitors, boiled in 50 μl Laemmli buffer (Bio-Rad) containing 2-mercaptoethanol, and analyzed by immunoblotting on Novex 10–20% Tris-Tricine gels (Life Technologies).

### Neuronal cell culture and lentivirus production

Primary hippocampal cultures were prepared from E19 rats as described previously and plated on glass coverslips coated with

poly-D-lysine (Guo *et al*, 2018). Lentivirus was packaged in HEK293FT cells as described (Guo *et al*, 2018).

## Immunofluorescence

After washing once with PBS, HeLa cells (2 days after transfection) and transduced primary hippocampal rat neurons (DIV3 + 4) were fixed for 10 min at room temperature (4% paraformaldehyde and 4% sucrose in PBS). Primary and secondary antibodies were diluted in GDB buffer (0.1% gelatin, 0.3% Triton X-100, 450 mM NaCl, 16 mM sodium phosphate pH 7.4). For visualizing the nucleus, cells were stained with DAPI (1:5,000 in PBS, 10 min, RT). After mounting the coverslips with Fluoromount™ Aqueous Mounting medium (Sigma), images were taken with LSM710 confocal microscope (Carl Zeiss, Jena) using a 63× oil immersion objective (NA 1.4).

## RNA isolation and quantitative RT–PCR

After 3 days of transfection, RNA isolation was conducted with the RNeasy- and QIAshredder kit (Qiagen) following the manufacturer's instructions. cDNA was generated using the TaqMan MicroRNA Reverse Transcription Kit (Applied Biosystems) with random hexamer primers according to the manufacturer's instructions. RT–qPCR was performed on the CFX384-Real-Time system (Bio-Rad) using following primers: CHCHD10 (Hs01369775_g1, Thermo Fisher), Mia40/CHCHD4 (Hs01027804_g1, Thermo Fisher), AIFM1 (Hs00377585_m1, Thermo Fisher), Erv1/GFER (Hs00193365_m1, Thermo Fisher), B2M (4326319E, Thermo Fisher), GAPDH (Hs02758991_g1, Thermo Fisher). Signals were normalized to GAPDH and B2M with the CFX Manager program (Bio-Rad) according to the $\Delta\Delta C_T$ method.

## Analysis of disulfide-bond formation

We used thiol-reactive 4-acetamido-4′-maleimidylstilbene-2,2′-disulfonic acid (AMS, Thermo Fisher) to analyze disulfide-bond formation following the protocol by (Gross *et al*, 2011). HeLa cells were lysed in RIPA buffer (137 mM NaCl, 20 mM Tris pH 7.5, 0.1% SDS, 10% glycerol, 1% Triton X-100, 0.5% deoxycholate, 2 mM EDTA) for 20 min on ice. After centrifugation (18,000 *g*, 15 min, 4°C), the supernatant was divided and incubated at room temperature or 95°C for 10 min with or without 15 mM dithiothreitol (DTT). Afterwards, proteins were precipitated with trichloroacetic acid (TCA). Here, one volume of a 8 M TCA stock solution was added to four volumes of protein sample, incubated at 4°C for 10 min, and centrifuged (18,000 *g*, 5 min, 4°C). After removing the supernatant, the pellet was washed with ice-cold acetone and again centrifuged (18,000 *g*, 5 min, 4°C). These washing steps were repeated twice, and the remaining pellet was dried at 95°C for 5–10 min. After acetone evaporation, the pellet was resolved in buffer (2% SDS, 100 mM Tris pH 8, 100 mM NaCl, 10 mM EDTA) and 10 mM AMS or distilled water was added. The samples were incubated for 60 min at 37°C in the dark. After adding 50 mM iodoacetic acid (IAA), Laemmli buffer (Bio-Rad) was added and the proteins were analyzed by immunoblotting using 12.5% SDS–PAGE gels.

## The paper explained

### Problem

Several mutations in CHCHD10 have been reported in familial and sporadic cases of amyotrophic lateral sclerosis (ALS), frontotemporal dementia (FTD), spinal muscular atrophy, and mitochondrial myopathy, but their mode of action is unclear. Since disease progression in mutation carriers is usually slow and penetrance is incomplete, some geneticists raised concerns, whether CHCHD10 mutations are truly pathogenic. CHCHD10 is a small protein localized to the intramembrane space of mitochondria. It is involved in organizing cristae morphology and has been linked to stability of mitochondrial DNA. Loss-of-function and gain-of-function pathomechanisms have been discussed. Several patient mutations, including R15L, are located in the proposed N-terminal mitochondrial targeting signal (MTS), but the mitochondrial import mechanism of CHCHD10 has not been carefully analyzed experimentally, although restoring mitochondrial import of CHCHD10 may be a therapeutic strategy.

### Results

We discovered a novel CHCHD10 mutation (Q108P) in a highly conserved residue within the coiled-coil-helix-coiled-coil-helix (CHCH) domain in a young ALS patient with aggressive disease progression and analyzed its pathogenicity in transfected heterologous cells and primary rat neurons. The Q108P mutation blocked mitochondrial import nearly completely suggesting a loss-of-function mechanism. Moreover, reduced CHCHD10 expression in heterologous and patient cells inhibited mitochondrial respiration. The R15L mutation had only a small effect on overall protein levels, but largely spared mitochondrial localization. Several other CHCHD10 variants reported in ALS/FTD patients showed diffuse cytoplasmic localization (C122R) or dot-like clustering within mitochondria (G58R, S59L, G66V, G66S, E127K) and reduced stability and/or expression (R15L, P23S, G58R, G66V, Q108P, Q108*, C122R). Mitochondrial import of CHCHD10 is predominantly driven by Mia40-dependent disulfide-bond formation in the CHCH domain rather than the putative N-terminal MTS. Overexpression of Mia40 strikingly boosts mitochondrial import of CHCHD10 Q108P.

### Impact

The identification of a novel CHCHD10 mutation resulting in aggressive ALS and a clear loss-of-function phenotype *in vitro* strongly supports the genetic role of CHCHD10 in ALS pathogenesis. This unusual mutation revealed Mia40-dependent mitochondrial import of CHCHD10 and suggests that activation of the Mia40-dependent mitochondrial import pathway could be a novel therapeutic strategy. Our data supports the pathogenicity of several previously uncharacterized CHCHD10 variants found in ALS/FTD patients via a loss-of-function mechanism (R15L, P23S, G58R, G66V, Q108P, Q108*, C122R) and/or gain-of-function mechanism (G58R, S59L, G66V, G66S, E127K).

## Quantitative analysis of respiration

Oxygen consumption rate (OCR) was measured using the Seahorse XF96 extracellular flux analyzer (Agilent). The day before, siCHCHD10 knockdown or control siRNA transfected HeLa cells were plated in growth medium in 96-well plates (Agilent). For OCR measurements, growth medium was replaced with pre-warmed XF assay medium (Agilent) supplemented with 10 mM glucose and 10 mM pyruvate, and cells were incubated at 37°C without $CO_2$ for 60 min. To measure OCR in patient lymphoblastoid cells, 96-well plates (Agilent) were coated with 30 μl of poly-D-lysine (50 μg/ml) in 0.1 M borate buffer (pH 8.5) for 2 h and washed twice with cell culture-grade water. One hour before the measurement,

lymphoblasts were plated ($1.1 \times 10^5$ cells/well) in pre-warmed XF assay medium (Agilent) and incubated at 37°C without $CO_2$. Oligomycin (final concentration 1 μM), FCCP (0.75 μM), and rotenone and antimycin A (10 μM each) were diluted with pre-warmed assay medium and loaded into injector ports. Assay cycles included 4 min of mixing, followed by 4 min of measurement.

**Expanded View** for this article is available online.

## Acknowledgements

We thank Hannelore Hartmann, Bettina Schmid, Harald Steiner, Johannes Trambauer, Matias Wagner, and Qihui Zhou for critical comments to the manuscript. This work was supported by NOMIS Foundation and the Hans und Ilse Breuer Foundation (D.E.), the Munich Cluster of Systems Neurology (SyNergy) (D.E.), the European Community's Health Seventh Framework Programme under Grant agreement no. 617198 [DPR-MODELS] (D.E.) and the general legacy of Mrs. Ammer (N.E.).

## Author contributions

DE, MTH, and CL designed the study and interpreted the results with additional help from NE and PW. CL performed experiments with help from MHS, LR, JG, MJ, and HR. JvdZ and CVB provided genetically characterized patient lymphoblasts. MTH identified the patient. DE, CL, and PW wrote the manuscript with input from all co-authors.

## Conflict of interest

The authors declare that they have no conflict of interest.

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
