## [Review Process File · EMBO Molecular Medicine]

A novel CHCHD10 mutation implicates a Mia40-dependent mitochondrial import deficit in ALS

Carina Lehmer, Martin H. Schludi, Linnea Ransom, Johanna Greiling, Michaela Junghänel, Nicole Exner, Henrick Riemenschneider, Julie van der Zee, Christine Van Broeckhoven, Patrick Weydt, Michael T. Heneka, Dieter Edbauer

Review timeline:

Submission date:	14 October 2017
Editorial Decision:	09 December 2017
Revision received:	19 March 2018
Editorial Decision:	12 April 2018
Revision received:	13 April 2018
Accepted:	18 April 2018

Editor: Céline Carret

Transaction Report:

1st Editorial Decision

09 December 2017

Thank you for the submission of your manuscript to EMBO Molecular Medicine. We have now heard back from the three referees whom we asked to evaluate your manuscript.

You will see from the copied reports below that all three referees, while finding the data of interest, also highlight significant concerns. While the technical aspects should be in principle the easiest to address, we would like to invite you to provide additional data of clinical and/or pre-clinical nature regarding this specific mutation. We also would like to encourage you to add further insights regarding the pathogenicity of the CHCHD10 variants identified up to now.

We would therefore welcome the submission of a revised version within three to six months for further consideration and would like to encourage you to address all the criticisms raised as suggested to improve conclusiveness and clarity. Please note that EMBO Molecular Medicine strongly supports a single round of revision and that, as acceptance or rejection of the manuscript will depend on another round of review, your responses should be as complete as possible.

Please also contact us as soon as possible if similar work is published elsewhere. If other work is published we may not be able to extend the revision period beyond the given time period.

I look forward to receiving your revised manuscript.

***** Reviewer's comments *****

Referee #1 (Comments on Novelty/Model System for Author):

Most experiments are based on overexpression. Analysis of patient cells would clarify the relevance of the findings

Referee #1 (Remarks for Author):

Lehmer et al describe a novel CHCHD10 mutation and show that, unlike previously identified mutations, it dramatically reduces mitochondrial. They show that the CHCH domain, within which the mutation is located, is required for mitochondrial import, rather than the previously predicted mitochondrial localisation sequence. Finally they show that CHCHD10 mitochondrial import is dependent on Mia40 and that overexpression of Mia40 is sufficient to redirect mutant CHCHD10 back to the mitochondria.

These findings provide novel insight into the action of a novel missense mutation in CHCHD10. However I have several issues with the manuscript in its current form. I could not find information on the number of biological replicates used for any experiment, no quantification is provided for any of the immunofluorescence or blotting data (which is the majority of the data) and no statistical analyses are performed, except in Figure 2. These issues make the results difficult to interpret. The mutation is described in a single case with no family history so it is difficult to be sure of its pathogenicity. Most experiments are based on overexpression of the mutant CHCHD10 in HeLa cells, which may not reflect the situation in patient cells.

Specific points:

Figure 1) Replicate data, and quantification of figures 1E, 1D and 1F would help to support the conclusion.

Figure 2) the title states CHCHD10 is necessary for mitochondrial repair. This does not appear to be the case as mitochondrial repair is reduced rather than abolished. The relevance is also unclear as the effect is observed by using a high level of knockdown, so may be different in patients with a heterozygous point mutation.

Figure 3) Again, presentation of images without quantification makes the data difficult to interpret. Especially when some mutations appear to have almost no expression of protein (e.g. Fig 3A delta CHCH). Perhaps the authors should try to select cells with equivalent expression if possible?

Figure 3C) This experiment does not allow a clear conclusion about the targeting ability of the different domains, as mitochondrial GFP targeting does not occur in any context.

Figure 4) A) statistics should be used for the RT-PCR data C) Quantification is needed D) Loading controls would be helpful.

Figure 5) Quantification is needed. C) This result is striking. Can the authors show the reciprocal experiment, that Mia40 overexpression leads to reduced Q108P in the cytoplasm or is it due to a more general increase in CHCHD10 levels?

Referee #2 (Remarks for Author):

Lehmer and colleagues report on a novel mutation in CHCHD10 that they discovered in a young patient with rapidly progressing ALS. The effects of the mutation (Q108P) on CHCHD10 biology and function were characterized in cell culture. From their experiments on this mutation the authors maintain that the CHCHD region regulates mitochondrial import and that loss of mitochondrial CHCHD10 causes impaired mitochondrial respiration. Mia40 overexpression can rescue the blocked mitochondrial import of CHCHD10-Q108P by enhancing disulfide bond formation in the CHCHD region, thus potentially being an avenue for therapeutics.

This manuscript has several strengths. While mutations in CHCHD10 have already been found to be linked to ALS and several other human diseases, they indeed discovered a novel mutation. The

overall study is generally a good example of how in theory a logical experimental design can take a pathogenic gene mutation to the level of pathological mode of action and therapeutic approaches. Also, the manuscript is concise and nicely readable.

The manuscript has some critical conceptual and specific weaknesses.

1. Conceptually, the first figure does not entirely support their conclusion that Q108P mutant inhibits mitochondrial import. In both HeLa cells (Fig. 1D) and neurons (Fig. 1F) there appears to be an appreciable amount of CHCHD10-Q108P localized to mitochondria. This interpretation is even supported by the western blot of subcellular fractions (Fig. 1E).
2. Figure 1 should have some quantification and appropriate statistical analyses of the cell imaging and western blotting showing the levels of CHCHD10-Q108P immunoreactivity in the subcellular compartments.
3. Also of conceptual importance they often do not account for basal levels of endogenous CHCHD10 in HeLa cells or neurons. For example, an indication of fold levels of exogenous (wild type and mutant) expression over basal levels would be appropriate in Figure 1.
4. The authors need to be certain that the CHCHD10-Q108P-HA expression vector is simply just as good as their other constructed expression vectors. It is stated that all vectors were sequenced, but was this just the coding regions or was the promoter sequenced too? The results should be replicated with a second CHCHD10-Q108P expression vector, perhaps with a different tag.
5. Figure 2 is nice because they show that they are indeed dealing with endogenous CHCHD10 and they can very effectively knock-down levels. However, in the presence of essentially complete knock-down of CHCHD10 the effects on mitochondrial respiration are modest. The graphs in 2B and E should have the same y-axis scaling. Also, the graphs in 2F and C should have the same y-axis scaling.
6. Figure 3 again supports the presence of some mitochondrial targeting of CHCHD10-Q108P mutant. Figure 3 should have some quantification. The WT-HA image is overexposed and saturated.
7. Figure 4. The values in 4A need to be defined and analyzed statistically. The AMS assay is clever. Are these observations dependent upon the high levels of CHCHD10-HA overexpression? How does this compare to endogenous (basal) CHCHD10?
8. Figure 5. The neuron cultures seem to be just not that good (healthy). Figure 1F fostered the similar impression. Some of the neurons shown are highly vacuolated with cytoplasmic holes (Fig 5B WT Mia40; Fig 1F Q108P) and others used for comparison are small neurons compared to others (Fig 5B Q108P Ctrl). The neuronal culture quality should be improved.

Referee #3 (Comments on Novelty/Model System for Author):

Equivocal. Does not employ relevant cells for the most part.

Referee #3 (Remarks for Author):

Comments:

This study by Lehmer et al describes a new clinical mutation in CHCHD10, a mitochondrial protein that has recently been identified as an ALS gene. The authors show that the Q108P mutation disrupts CHCHD10 mitochondrial import and diminishes its expression. They further show that CHCHD10 import is independent of its mitochondrial leader and is mediated by Mia40, whose overexpression can rescue the expression and import defects of the Q108P mutant. CHCHD10 knockout cells also exhibit mitochondrial respiration defects suggestive of an LOF mechanism.

This is an interesting study that clearly documents mitochondrial targeting defects of the CHCHD10 Q108P mutant, which appears to have unique properties. The Mia40 data are also impressive and interesting. My main concern is that this study does not go far enough in light of previous work on ALS-associated CHCHD10 mutations, including two studies published back-to-back in HMG earlier this month. I think the authors would need to go further in light of these other studies.

Specific Comments

1. The authors strongly imply that the Q108P mutant is unstable. However there are no experiments to directly test stability.
2. Most of the experiments employ transient expression which can lead to vastly different transgene expression levels that probably greatly exceed those of the endogenous gene. Retroviral vectors and/or inducible lines on a CHCHD10-null background would have been preferable.
3. The Mia40 data is interesting and relatively convincing but raises several obvious questions. Do CHCHD10 and Mia40 interact? Do Q108P or CHCH domain mutations affect binding? Does Mia40 increase CHCHD10 protein half life?
4. Fig. 5 would benefit from additional controls. Ideally the authors would identify and test a Mia40 mutant that does not bind CHCHD10 and/or a CHCHD10 mutant that does not bind Mia40.

Minor comments:

1. How are the authors able to observe such strong CHCHD10 staining in Fig 1D is unclear given that Western signal is so low. If this is due to a gain enhancement, it should be made clear to the reader.
2. IF images in Fig. 3C are too dim.

Author correspondence

21 February 2018

Here is a quick update on our revisions for the CHCHD10 Q108P manuscript:

- We successfully addressed all the technical and biochemical questions. This new data further confirms the import deficit of the CHCHD10 Q108P mutant by rigorous quantification and statistics. Furthermore, we can now show that Q108 stability is decreased and can be rescued by expression of Mia40.
- We successfully generated embryonic fibroblasts from our Q108P knockin mice to analyze respiration and localization. Unfortunately, even the wildtype fibroblasts don't express any CHCHD10 at this stage, which makes the comparison meaningless.
- We analyzed Q108* lymphoblasts from a Belgian FTD patient. These cells have 50% reduction in mRNA and protein and also show reduced spare capacity in respiration assays, which nicely supports the loss-function mechanism for CHCHD10 pathogenesis.
- We examined all published CHCHD10 missense variants. Several mutations also show reduced expression levels (P23S, G58R, G66V). Some mutants within the hydrophobic domain lead to clustering of CHCHD10 in small punctae within mitochondria (S59L, G66V, G66S) suggesting CHCHD10 mutations can cause ALS/FTD via two distinct pathomechanisms.
- In addition, we characterized two novel mutations in the CHCH domain that we found in an ALS exome database (ALSdb, C122R and E127K). Importantly, C122R affects one of the critical cysteines within the CHCH domain and blocks mitochondrial import similar to Q108P. E127K also shows reduced mitochondrial import and a dot-like expression pattern like the mutants in the hydrophobic domain. These new data strongly support the role of the CHCH domain for mitochondrial import of CHCHD10 via Mia40.
- Our Q108P patient is still alive and eager to give blood to generate lymphoblasts (the Russian physicians cannot collect fibroblasts or do any experiments themselves). However, we are struggling since Christmas to organize the shipment. It's theoretically possible, but it's an administrative nightmare because shipping blood across the Russian border is illegal without a special license that is very hard to obtain. We failed with Fedex and are now trying with DHL. Thus, getting data from our own patient will take another 2-3 months, but it is not guaranteed that we will receive vital blood cells for transformation to lymphoblasts.
- Recently a third CHCHD10 paper appeared in HMG (Brockman et al 2018) that also supports a loss of function mechanism for R15L and G66V. What would you like us to do? We can submit the manuscript with the Q108* lymphoblasts and the additional CHCHD10 mutants within 2-3 weeks or keep pushing for the Q108P cells and submit in about 2-3 months.

Thank you very much for your consideration.

Editor's response

22 February 2018

I am very sorry to hear about your difficulties, first with the KI mouse then with the blood of the Russian patient. This is quite unfortunate.

I have discussed with my colleagues and we feel that at this stage, we should not try and waste more time running after an authorisation that even if it comes may not result in viable cells to pursue the experiments. Along with all the data you're adding and the new paper corroborating your findings I think we should move forward.

Therefore I'd like you to submit your paper as soon as it's ready. Please make sure to provide this information in your rebuttal letter.

1st Revision - authors' response

19 March 2018

Point-by-point response to the referees

Referee #1

(Comments on Novelty/Model System for Author)

Most experiments are based on overexpression. Analysis of patient cells would clarify the relevance of the findings

We agree with the reviewer about the relevance of patient cells to confirm our findings. We analyzed cells from a patients with Q108* mutation, which results in ~50% reduction of mRNA and proteins levels due to non-sense mediated decay. In patient lymphoblasts spare respiratory capacity is reduced as in Hap1 and HeLa knockdown cells.

(Remarks for Author)

Lehmer et al describe a novel CHCHD10 mutation and show that, unlike previously identified mutations, it dramatically reduces mitochondrial. They show that the CHCH domain, within which the mutation is located, is required for mitochondrial import, rather than the previously predicted mitochondrial localisation sequence. Finally they show that CHCHD10 mitochondrial import is dependent on Mia40 and that overexpression of Mia40 is sufficient to redirect mutant CHCHD10 back to the mitochondria.

These findings provide novel insight into the action of a novel missense mutation in CHCHD10. However I have several issues with the manuscript in its current form. I could not find information on the number of biological replicates used for any experiment, no quantification is provided for any of the immunofluorescence or blotting data (which is the majority of the data) and no statistical analyses are performed, except in Figure 2. These issues make the results difficult to interpret. The mutation is described in a single case with no family history so it is difficult to be sure of its pathogenicity. Most experiments are based on overexpression of the mutant CHCHD10 in HeLa cells, which may not reflect the situation in patient cells.

We agree with the reviewer about the need for more quantitative data. We added quantification and statistics for the mitochondrial import of CHCHD10 mutants based on biochemical fractionation for all key experiments (new Fig. 1D/E, Fig 3B/C, Fig. 4C/D, Fig. 6C/D). Full statistical information is provided in all revised figure legends.

The family history of the patient is negative and we do not have DNA from the parents to establish a de novo mutation. Interestingly, FUS de novo mutations are a common cause of early onset ALS without family history (Hübers et al, Neurobiol Aging 2015). Therefore, we suspect a similar mechanism in this case. Moreover, a Q108* mutation found in FTD cases (Perrone et al. 2017) leads to ~50% reduced CHCHD10 expression and reduced spare respiratory capacity in patient lymphoblasts strongly supporting a loss of function component in CHCHD10 ALS/FTD (new Fig. 2G-I). Importantly, the ALSdb contains two additional mutations within the CHCHD10 (C122R and E127K) that are expected to disrupt the CHCH domain suggesting this pathomechanism is common. C122 is one of the critical cysteine residues in the CHCH domain. Indeed, CHCHD10 C122R shows a diffuse cytoplasmic localization and reduced expression levels like Q108P indicating a similar

pathomechanism (new Fig. 4). In addition, the patient variants P23S, G58R and G66V also show reduced expression. Surprisingly, CHCHD10 E127K shows punctuate expression similar to the pathogenic S59L mutation first described in Bannwarth et al (2014). We now show a similar pattern for several other known and novel CHCHD10 variants (G58R, G66V, G66S) suggesting these mutations may cause gain-of-function toxicity.

Although many of our findings are from transfected cells, expression levels are comparable to endogenous CHCHD10 (see extended Fig. 1D). In addition, we replaced Fig 1F and 6B with lentivirally transduced neurons, which show the same cytoplasmic mislocalization of Q108P as previous transfection experiments.

Specific points:

Figure 1) Replicate data, and quantification of figures 1E, 1D and 1F would help to support the conclusion.

As indicated above, we added quantification for the mitochondrial and cytosolic fractionation (new Fig. 1E). From regular confocal images, direct co-localization analysis of CHCHD10 and mitochondrial markers is error-prone because cytosolic proteins do not show a negative image of mitochondria at the confocal resolution. Manual quantification of “diffuse” vs “mitochondrial” cells would not give more robust data than showing the images directly. Nevertheless, we included representative line scans of immunofluorescence pictures of WT, Q108P and R15L CHCHD10 showing that only Q108P is present in ATP5A1 negative areas (new Fig. EV1B).

Figure 2) the title states CHCHD10 is necessary for mitochondrial repair. This does not appear to be the case as mitochondrial repair is reduced rather than abolished. The relevance is also unclear as the effect is observed by using a high level of knockdown, so may be different in patients with a heterozygous point mutation.

This is a valid point. We now include respiration data from patient lymphoblasts carrying a heterozygous Q108* mutation (new Fig. 2G-I), which shows 50% reduced mRNA levels and reduced proteins levels due to nonsense-mediated decay. These cells show significantly reduced spare capacity in the respiration analysis. Thus, we changed the legend title to “Partial loss of CHCHD10 reduces respiratory spare capacity”.

Figure 3) Again, presentation of images without quantification makes the data difficult to interpret. Especially when some mutations appear to have almost no expression of protein (e.g. Fig 3A delta CHCH). Perhaps the authors should try to select cells with equivalent expression if possible?

As for Figure 1, we now added quantification for the biochemical fractionation, which shows significantly reduced mitochondrial import of CHCHD10 lacking the CHCH domain or containing the Q108* and Q108P mutations (extended Fig 3B,C). The original figure contained images reflecting the average expression levels, but the reviewer is correct that cells with similar expression levels are more informative. We provide this new data in the revised Figure 3A. Consistent with the fractionation experiments, Δ CHCH is diffusely localized even in high-expressing cells.

Figure 3C) This experiment does not allow a clear conclusion about the targeting ability of the different domains, as mitochondrial GFP targeting does not occur in any context.

We agree with the reviewer. Unfortunately, neither domain is sufficient to import of GFP into mitochondria. However, fusing GFP to the N- or C-terminus of full-length CHCHD10 also blocked mitochondrial import of wild-type CHCHD10 (data not shown) indicating that the CHCH domain-mediated import mechanism may not be compatible with large proteins. Since GFP-fusion is a standard assay to analyze targeting motif we would like to include the negative data anyway. We clearly state the limitation in the revised manuscript.

Figure 4) A) statistics should be used for the RT-PCR data C) Quantification is needed D) Loading controls would be helpful.

We apologize for the missing statistics. We now provide statistics for the mRNA levels and quantifications for the protein levels (revised Fig 5A, new Figure 5C). In addition, we provide a loading control in the revised Fig. 5E.

Figure 5) Quantification is needed. C) This result is striking. Can the authors show the reciprocal experiment, that Mia40 overexpression leads to reduced Q108P in the cytoplasm or is it due to a more general increase in CHCHD10 levels?

This an excellent suggestion. Quantification of the biochemical fraction confirms the strong effect of Mia40 expression on CHCHD10 Q108P import (revised Fig 6C,D). The cytoplasmic levels of Q108P are indeed increased as well, but we attribute this to the partial cytoplasmic localization of Mia40 upon overexpression (new Fig. EV4C)

Referee #2

(Remarks for Author)

Lehmer and colleagues report on a novel mutation in CHCHD10 that they discovered in a young patient with rapidly progressing ALS. The effects of the mutation (Q108P) on CHCHD10 biology and function were characterized in cell culture. From their experiments on this mutation the authors maintain that the CHCHD region regulates mitochondrial import and that loss of mitochondrial CHCHD10 causes impaired mitochondrial respiration. Mia40 overexpression can rescue the blocked mitochondrial import of CHCHD10-Q108P by enhancing disulfide bond formation in the CHCHD region, thus potentially being an avenue for therapeutics.

This manuscript has several strengths. While mutations in CHCHD10 have already been found to be linked to ALS and several other human diseases, they indeed discovered a novel mutation. The overall study is generally a good example of how in theory a logical experimental design can take a pathogenic gene mutation to the level of pathological mode of action and therapeutic approaches. Also, the manuscript is concise and nicely readable.

We thank the reviewer for the kind summary of our work.

The manuscript has some critical conceptual and specific weaknesses.

1. Conceptually, the first figure does not entirely support their conclusion that Q108P mutant inhibits mitochondrial import. In both HeLa cells (Fig. 1D) and neurons (Fig. 1F) there appears to be an appreciable amount of CHCHD10-Q108P localized to mitochondria. This interpretation is even supported by the western blot of subcellular fractions (Fig. 1E).

We agree that the biochemical experiments show low amounts of CHCHD10 Q108P in the mitochondrial fraction. We cannot distinguish, whether this represents actual residual import or cytosolic contamination during the fractionation procedure and rephrased our claim accordingly. Since CHCHD10 Q108P levels are reduced in the mitochondrial but not the cytosolic fraction (new Fig 1C), the most likely explanation for the reduced expression and stability is reduced mitochondrial import, which is fully supported by the truncation studies (Fig. 3) and the Mia40 knockdown and overexpression data (Fig 5 and 6).

Due to the resolution limit of confocal microscopy cytosolic proteins still colocalize with mitochondrial staining as shown below for endogenous cytosolic mTOR and ATP5A1. However, representative line scans of CHCHD10 and ATP5A1 show that WT and R15L intensity correlates with mitochondrial staining, while Q108P intensity does not (new Fig. EV1B). Thus, we now provide quantification of all biochemical fractionation experiments to allow rigorous conclusions on mitochondrial import.

2. *Figure 1 should have some quantification and appropriate statistical analyses of the cell imaging and western blotting showing the levels of CHCHD10-Q108P immunoreactivity in the subcellular compartments.*

Due to the resolution limit of confocal microscopy, we rather provide quantification of the biochemical fractionation, which shows less than 20% residual CHCHD10 Q108P in mitochondria while cytosolic levels are comparable to CHCHD10 WT (new Fig. 1D,E).

3. *Also of conceptual importance they often do not account for basal levels of endogenous CHCHD10 in HeLa cells or neurons. For example, an indication of fold levels of exogenous (wild type and mutant) expression over basal levels would be appropriate in Figure 1.*

We provide this important data in the revised Fig 1D. The wild type CHCHD10 is expressed at similar levels as the endogenous CHCHD10. We did not detect “replacement” of endogenous CHCHD10 by overexpressed wild type or mutant CHCHD10, which argues against dominant negative effects of the tested variants.

4. *The authors need to be certain that the CHCHD10-Q108P-HA expression vector is simply just as good as their other constructed expression vectors. It is stated that all vectors were sequenced, but was this just the coding regions or was the promoter sequenced too? The results should be replicated with a second CHCHD10-Q108P expression vector, perhaps with a different tag.*

We performed several experiments to exclude lower expression of the Q108P mutant. First, sequencing of the human ubiquitin promoter showed no differences in the wild type, Q108P and R15L vectors (data not shown). Second, we generated another CHCHD10 expression constructs with optimized codon usage using gene synthesis (sequence provided in new Fig. EV2A) and introduced R15L, Q108P and a series of other mutations (new Fig. 4). These constructs show overall higher expression levels, but the Q108P and R15L levels are still reduced compared to wild type also in this background (new Fig. EV2B). Most importantly, Q108P also impairs mitochondrial import in this vector, which supports a bona fide protein-based pathomechanism for the R15L and Q108P mutation in our patient (new Fig EV2C and 4C/D).

5. *Figure 2 is nice because they show that they are indeed dealing with endogenous CHCHD10 and they can very effectively knock-down levels. However, in the presence of essentially complete knock-down of CHCHD10 the effects on mitochondrial respiration are modest. The graphs in 2B and E should have the same y-axis scaling. Also, the graphs in 2F and C should have the same y-axis scaling.*

We agree that siRNA and knockout alter CHCHD10 levels more drastically than it is expected for heterozygous Q108P mutation. Since we could not obtain cell from this patient in time for this revision, we analyzed lymphoblasts from an FTD patient with heterozygous Q108* mutation. Compared to controls, CHCHD10 mRNA and protein levels are reduced by ~50%. These cells also show reduced spare capacity in the respiration assay. CHCHD10 is not a direct component of the respiratory chain and therefore modest effects on respiration are expected. The reduced spare respiratory capacity in knockdown and patient cells suggests that CHCHD10 still contributes to efficient respiration.

The y-axis shows respiration normalized the total protein content (Fig 2B/E) or cell number (Fig 2H), but the respiration rate likely differs between different cell types. We clearly state this in the revised legends to avoid confusion.

6. *Figure 3 again supports the presence of some mitochondrial targeting of CHCHD10-Q108P mutant. Figure 3 should have some quantification. The WT-HA image is overexposed and saturated.*

As for Figure 1, we now added quantification for the biochemical fractionation, which shows significantly reduced mitochondrial import of CHCHD10 lacking the CHCH domain or containing the Q108* and Q108P mutations (extended Fig 3B,C). In fact, the residual import is only visible because we show a slightly overexposed blot. Thus, the quantification likely underestimates the true import deficit of Q108P.

7. *Figure 4. The values in 4A need to be defined and analyzed statistically. The AMS assay is clever. Are these observations dependent upon the high levels of CHCHD10-HA overexpression? How does this compare to endogenous (basal) CHCHD10?*

We analyzed the expression data statistically as requested (revised/new Fig. 5A/B/C). CHCHD10-HA transfection results in expression levels comparable to endogenous expression (extended in Fig. 1D).

Endogenous CHCHD10 behaves like CHCHD10-HA in the AMS assay consistent with disulfide-bond formation under basal conditions (new Fig EV4B).

8. *Figure 5. The neuron cultures seem to be just not that good (healthy). Figure 1F fostered the similar impression. Some of the neurons shown are highly vacuolated with cytoplasmic holes (Fig 5B WT Mia40; Fig 1F Q108P) and others used for comparison are small neurons compared to others (Fig 5B Q108P Ctrl). The neuronal culture quality should be improved.*

The criticism is justified. We repeated the experiments in a better batch of neurons and show new images with neurons of similar size (revised Fig. 1F and 6B).

Referee #3

*(Comments on Novelty/Model System for Author)
Equivocal. Does not employ relevant cells for the most part.*

We respectfully disagree about the relevance of our cellular models because mitochondrial import is highly conserved between different cell types. Importantly, we analyzed CHCHD10 in primary neurons, probably the most relevant cell type for ALS/FTD. Moreover, we now added data from patient lymphoblasts containing a Q108* mutation, which fully support the respiration data from HAP1 an HeLa cells (extended Fig. 2). Thus, we report matching observations in patient lymphoblasts, heterologous cells and primary neurons.

*(Remarks for Author)
Comments:*

This study by Lehmer et al describes a new clinical mutation in CHCHD10, a mitochondrial protein that has recently been identified as an ALS gene. The authors show that the Q108P mutation disrupts CHCHD10 mitochondrial import and diminishes its expression. The further show that CHCHD10 import is independent of its mitochondrial leader and is mediated by Mia40, whose overexpression can rescue the expression and import defects of the Q108P mutant. CHCHD10 knockout cells also exhibit mitochondrial respiration defects suggestive of an LOF mechanism.

This is an interesting study that clearly documents mitochondrial targeting defects of the CHCHD10 Q108P mutant, which appears to have unique properties. The Mia40 data are also impressive and interesting. My main concern is that this study does not go far enough in light of previous work on ALS-associated CHCHD10 mutations, including two studies published back-to-back in HMG earlier this month. I think the authors would need to go further in light of these other studies.

We provide all data specifically requested by the reviewers. In addition, we characterized 2 novel mutations in the CHCH domain and all known CHCHD10 missense variants reported in ALS and FTD patients (new Fig. 4 and EV3). Our data supports loss of function mechanisms by reduced stability/mitochondrial import for R15L, P23S, G58R, G66V, Q108P, Q108*, and the novel C122R mutation (new Fig. 4, Fig 3B-C and EV1C-E). Several mutations cause clustering of CHCHD10 in mitochondria (G58R, S59L, G66V, G66S, and the novel E127K), which may support a gain-of-function mechanism. Other reported mutations had no obvious effect on CHCHD10 localization or stability suggesting they may be benign variants. Systematic comparison of all known CHCHD10 variants for the first time strongly supports the role of CHCHD10 in ALS/FTD pathogenesis by interfering with mitochondrial function.

Specific Comments

1. The authors strongly imply that the Q108P mutant is unstable. However there are no experiments to directly test stability.

We followed this very reasonable suggestion and now quantified the half-life time of CHCHD10 wild-type and mutants using cycloheximide treatment, which fully confirms our assumption that CHCHD10 Q108P is less stable in cells than wild type protein (new Fig. EV1C-E).

2. Most of the experiments employ transient expression which can lead to vastly different transgene expression levels that probably greatly exceed those of the endogenous gene. Retroviral vectors and/or inducible lines on a CHCHD10-null background would have been preferable.

It is true that transient expression often exaggerates effects due to abnormally high expression levels. However, immunoblots show that transient transfection of CHCHD10 wild type results in expression levels comparable to endogenous CHCHD10 (extended Fig. 1D). Exogenous expression had no effect on endogenous levels, arguing against dominant negative effects in our experiments. In addition, lentiviral expression of CHCHD10 constructs in primary neurons shows similar import deficits for the Q108P mutant (revised Fig. 1F, 6B).

3. The Mia40 data is interesting and relatively convincing but raises several obvious questions. Do CHCHD10 and Mia40 interact? Do Q108P or CHCH domain mutations affect binding? Does Mia40 increase CHCHD10 protein half life?

These are important points. We confirmed interaction of CHCHD10 and Mia40 using co-immunoprecipitation (new Fig. EV4A). Since interaction with the oxidoreductase Mia40 is likely transient, we had to employ Mia40 overexpression to detect interaction. Under these conditions CHCHD10 Q108P is stabilized and co-immunoprecipitation of Mia40 is only slightly reduced. However, deleting the CHCH domain in CHCHD10 nearly abolished binding to Mia40. Moreover, Mia40 expression significantly increase protein stability of CHCHD10 WT, Q108P and R15L in CHX experiments (new Fig. EV4D,E).

4. Fig. 5 would benefit from additional controls. Ideally the authors would identify and test a Mia40 mutant that does not bind CHCHD10 and/or a CHCHD10 mutant that does not bind Mia40.

Peleh et al. eLife 2016 described two Mia40 point mutations in yeast that disrupt catalytic activity (aa 296-298 CPC to SPS, in human aa 66-68) or the hydrophobic binding pocket for its substrates (aa 315-318 FSCE to ESCE, in human aa 85-88) and we introduced the same mutations into human Mia40. These mutants do not recruit Q108P into mitochondria, but unfortunately, they are not localized to mitochondria themselves, which precludes a meaningful analysis. The different behavior compared to the yeast mutants may be due to the different structure of Mia40, which contains a transmembrane domain for mitochondrial anchoring in yeast, but not in higher eukaryotes. Designing new mutants with better properties is beyond the scope of this revision.

The Q108P mutation is predicted to break the α -helical structure of the CHCH domain. We therefore additionally tested a Q108G variant because glycine is a powerful helix-breakers like proline. Indeed, Q108G impairs mitochondrial import similar to Q108P (see below) suggesting that α -helical structure of CHCHD10 is critical for Mia40 binding. Moreover, the identical phenotype of Q108P

and C122R strongly suggests that Mia40-mediated disulfide-bond formation is critical for mitochondrial localization of CHCHD10.

Minor comments:

1. How are the authors able to observe such strong CHCHD10 staining in Fig 1D is unclear given that Western signal is so low. If this is due to a gain enhancement, it should be made clear to the reader.

All images in a panel are taken with the same microscope settings and without postprocessing of the intensity. We replaced this figure with new images due to concerns about neuronal health raised by reviewer #2. A high expressing cell is shown at the same microscope settings. Lower expressing cells also show no mitochondrial localization but are harder to see in these settings. We clearly state this in revised legend. Having no clear mitochondrial signal even in the highest-expressing cells suggests that import of Q108P is not due to overexpression.

2. IF images in Fig. 3C are too dim.

We replaced the images as requested.

2nd Editorial Decision

12 April 2018

Thank you for the submission of your revised manuscript to EMBO Molecular Medicine. We have now received the enclosed reports from the referees that were asked to re-assess it. As you will see the reviewers are now supportive and I am happy to inform you that we will be able to accept your manuscript pending final editorial amendments.

Please submit your revised manuscript within two weeks. I look forward to seeing a revised form of your manuscript as soon as possible.

I look forward to reading a new revised version of your manuscript as soon as possible.

***** Reviewer's comments *****

Referee #1 (Remarks for Author):

The manuscript has been improved by the additional data and quantification.

Referee #2 (Remarks for Author):

The authors have done a very nice and dutiful job at revising the manuscript. I have no additional suggestions.

Referee #3 (Remarks for Author):

None. Reviews were addressed though small concerns remain over impact/novelty in light of the earlier studies.

2nd Revision - authors' response

13 April 2018

Thank you very much for forwarding us the positive reviews. We corrected the manuscript as requested and are confident that it is now suitable for publication in EMBO Molecular Medicine.

Corresponding Author Name: Dieter Edbauer

Journal Submitted to: EMBO Mol Med

Manuscript Number: EMM-2017-08558